



# Sensitivity of stratospheric water vapour to variability in tropical tropopause temperatures and large-scale transport

Jacob W. Smith[1], Peter H. Haynes[2], Amanda C. Maycock[3], Neal Butchart[4], and Andrew C. Bushell[5]

[1]Department of Plant Sciences, University of Cambridge, Cambridge, UK
[2]Department of Applied Mathematics and Theoretical Physics, University of Cambridge, Cambridge, UK
[3]School of Earth and Environment, University of Leeds, Leeds, UK
[4]Met Office Hadley Centre, Exeter, UK
[5]Met Office, Exeter, UK

**Correspondence:** Jacob W. Smith (jws52@cam.ac.uk)

**Abstract.** Concentrations of water vapour entering the tropical lower stratosphere are primarily determined by conditions that air parcels encounter as they are transported through the tropical tropopause layer (TTL). Here we quantify the relative roles of variations in TTL temperatures and transport in determining seasonal and interannual variations of stratospheric water vapour. Following previous studies, we use trajectory calculations with the water vapour concentration set by the Lagrangian Dry Point along trajectories. To isolate the roles of transport and temperatures, the Lagrangian Dry Point calculations are modified by time-shifting temperatures relative to transport, and vice versa, with the shift made by years to investigate interannual variations and by months to investigate seasonal variations. Both ERA-Interim reanalysis data for the 1999-2009 period and data generated by a chemistry-climate model (UM-UKCA) are investigated. Variations in temperatures, rather than transport, dominate interannual variability, typically explaining more than 70 % of variability, including individual events such as the 2000 stratospheric water vapour drop. Similarly seasonal variation of temperatures, rather than transport, is shown to be the dominant driver of the annual cycle in lower stratospheric water vapour concentrations in both model and reanalysis, but it is also shown that seasonal variation of transport plays an important role in reducing the seasonal cycle maximum (reducing the annual range by about 30 %).

The quantitative role in dehydration of sub-seasonal and sub-monthly Eulerian temperature variability is also examined by using time-filtered temperature fields in the trajectory calculations. Sub-monthly temperature variability reduces annual mean water vapour concentrations by 40 % in the reanalysis calculation and 30 % in the model calculation. These results indicate that, whilst capturing seasonal and interannual variation of temperature is a major factor in modelling realistic stratospheric water vapour concentrations, simulation of seasonal variation of transport and of sub-seasonal and sub-monthly temperature variability are also important and cannot be ignored.

## 1 Introduction

Water vapour concentrations in the stratosphere are very low compared to those in the troposphere but have significant radiative and chemical impacts on the global climate system. For example, stratospheric water vapour changes on decadal



to multi-decadal timescales contribute to radiative forcing and surface temperature trends (Forster and Shine, 1999, 2002), to stratospheric temperature trends (Forster and Shine, 1999; Maycock et al., 2014), and can affect tropospheric circulation
through stratosphere-troposphere dynamical coupling (Maycock et al., 2013). Stratospheric water vapour also alters the catalytic cycles that destroy ozone through production of $HO_x$ radicals in the lowermost and upper stratosphere. It is therefore important to understand the processes that control stratospheric water vapour concentrations and to assess whether or not those processes are adequately represented in the models used for climate projections.

The basic process controlling stratospheric water vapour entry was identified by Brewer (1949). The dominant pathway by
which air enters the stratosphere is via transport from the relatively moist upper troposphere, through the tropical tropopause layer (TTL) (Fueglistaler et al., 2009) into the lower stratosphere (Holton et al., 1995). The low temperatures in the TTL correspond to low saturation mixing ratios, and air is 'freeze dried' as it is transported to the lower stratosphere resulting in very low water vapour concentrations (a few ppmv). Precise concentrations depend on the lowest saturation mixing ratios, which depend primarily on temperature but also pressure, sampled by air parcels as they pass through the TTL.

Due to the three-dimensional pathways traced by air parcels, latitudinal and longitudinal variation in the TTL, as well as the vertical variation, are important in determining stratospheric water vapour concentrations (Holton and Gettelman, 2001; Hatsushika and Yamazaki, 2003; Jensen and Pfister, 2004; Bonazzola and Haynes, 2004; Fueglistaler et al., 2004, 2005). This naturally motivates a Lagrangian perspective, in which time histories of saturation mixing ratio of air parcels moving from the troposphere to stratosphere must be considered, with the water vapour concentration entering the stratosphere set by the
minimum saturation mixing ratio encountered, commonly known as the Lagrangian Dry Point (LDP) (Liu et al., 2010). Average concentrations can be estimated by following large numbers of air parcel trajectories, using large-scale wind fields from reanalysis data or from model output, and making simple assumptions about the dehydration process, e.g. that water vapour concentrations adjust rapidly to the local saturation mixing ratio. Such calculations provide useful leading-order insight into seasonal and interannual variation in stratospheric water vapour (e.g. Fueglistaler and Haynes, 2005; Liu et al., 2010; Schoeberl
and Dessler, 2011), whilst accepting that important details such as convective-scale transport, mixing and microphysical details that determine particle formation, sedimentation and potential sublimation, are omitted.

The nature of the dehydration process means that Eulerian time variability of the TTL temperature field is potentially important in setting average water vapour concentrations. This has been noted by previous authors in the context of equatorial Kelvin waves (Fujiwara et al., 2001), of gravity waves and other equatorial waves on sub-monthly timescales (Kim and
Alexander, 2015), and of temporary cooling of the equatorial lower stratosphere associated with stratospheric sudden warmings (Takashima et al., 2010; Evan et al., 2015; Tao et al., 2015). Furthermore, Fueglistaler et al. (2013) showed explicitly from LDP calculations, that sub-monthly variations in TTL temperatures have a much larger effect on water vapour concentrations than longitudinal variations in temperature. This conclusion is slightly surprising bearing in mind that the picture of TTL dehydration which has emerged over the last 20 years has tended to emphasise the sampling of the longitudinal variation of temperatures as particularly important in setting water vapour concentrations. Kim and Alexander (2015) showed that
sub-monthly time variability of temperatures is underestimated in reanalyses and suggested this is a limitation of dehydration estimates based on such datasets.



Strong interannual variability of water vapour has been observed over the last three decades and is clearly associated with corresponding interannual variability in TTL temperatures (Randel et al., 2004; Fueglistaler and Haynes, 2005). However,

along with the interannual variability in temperatures, there are also substantial interannual variations of upwelling velocities in the TTL, and hence transport timescales (Ploeger and Birner, 2016; Abalos et al., 2012). An important aspect of the Lagrangian description of TTL dehydration is that the water vapour concentrations entering the stratosphere depend on both the temperature field and transport pathways through the TTL, since it is the combination of the two that determines saturation mixing ratios along trajectories. However, the relative importance of the two effects for variations in water vapour concen-

trations in the tropical lower stratosphere is not clear. For the observed strong step-wise drop in tropical lower stratospheric water vapour concentrations in late 2000 (Randel et al., 2006; Brinkop et al., 2016), Hasebe and Noguchi (2016) undertook an LDP-based study of variability and concluded the drop was caused by a combination of both anomalously cold temperatures and a modification of three-dimensional transport pathways due to weaker horizontal confinement of air masses encircling the Asian summer monsoon. However, their analysis did not include any quantitative separation of the two effects.

Whilst there has been progress in understanding the processes that affect stratospheric water vapour variability and trends, some important questions remain. In particular, the requirements for global climate models to capture the distribution of stratospheric water vapour, its seasonal and interannual variations and any long-term trends remain unclear. Many current models have systematic biases in TTL temperatures (Kim et al., 2013; Hardiman et al., 2015) and this likely causes biases in stratospheric water vapour concentrations (Keeble et al., 2020). Global models are also likely to be limited by their representation of

small-scale dynamical and microphysical processes. Thus, for example, if future changes in stratospheric water vapour were to be driven in part by changes in ice injection into the stratosphere by convective clouds (e.g. Dessler et al., 2016) or by changes in temperature fluctuations associated with gravity waves (e.g. Kim and Alexander, 2015), then global models might fail to capture such changes. The two aspects of dehydration highlighted above – the combined effects of transport and temperatures and the role of time variability in temperatures – are both likely to be important, but have not yet been clearly quantified. This

motivates the following questions:

– Is variation in temperatures or in transport more important in determining seasonal, interannual and longer term variations in water vapour?

– What is the role of dynamical and hence temperature variability on different time scales in setting water vapour concentrations?

These two questions will be addressed in the following sections using a novel Lagrangian approach that explicitly separates temperature and transport effects. Section 2 describes the datasets and methods used for the calculations. Section 3 describes the results of isolating the roles of temperature and large-scale transport in interannual and seasonal variability of water vapour entering the stratosphere, as well as the impact of sub-seasonal temperature variability, using reanalysis data. Section 4 presents corresponding results for a climate model. Overall conclusions will then be presented in Sect. 5.





## 2 Methods and Data

### 2.1 Trajectory and LDP calculations

We calculated Lagrangian Dry Points and water vapour concentrations using estimates of saturation mixing ratios evaluated across large sets of back trajectories. The back trajectories were calculated using the OFFLINE trajectory model (Methven, 1997; Liu, 2009) taking as input winds, temperatures and diabatic heating rates either from reanalysis data (Sect. 2.2) or from a global chemistry-climate model (Sect. 2.3).

The back trajectories were initialised once a month in the lower stratosphere (see Sect. 2.2 and 2.3 for details) and distributed every 2° in longitude and latitude over the tropical region between 30° N and 30° S, corresponding to 5580 trajectories per initialisation date. Each back trajectory was followed for a maximum of 1 year or until it reached the troposphere defined by $\theta < 340$ K and a saturation mixing ratio greater than 1000 ppmv. The fraction of trajectories that reach the troposphere within 1 year is typically about 90 %, consistent with previous studies (Liu et al., 2010; Fueglistaler et al., 2013). These are often referred to as 'troposphere-to-stratosphere' or 'TST-trajectories'. For each back trajectory, the time series of temperature and pressure along the trajectory is used to calculate a corresponding time series of saturation mixing ratio. The Lagrangian Dry Point (LDP) is the location along the trajectory where the saturation mixing ratio is minimum. This gives a water vapour estimate at a space time location $(\boldsymbol{x}_0, t_0)$ which can be denoted by $\text{SMR}_{\text{LDP}}(\boldsymbol{x}_0, t_0)$, representing the saturation mixing ratio evaluated at the LDP along the back trajectory starting at $(\boldsymbol{x}_0, t_0)$. The estimate for the tropical water vapour concentration at time $t_0$ is then $\overline{\text{SMR}_{\text{LDP}}}(t_0) = \langle \text{SMR}_{\text{LDP}}(\boldsymbol{x}_0, t_0) \rangle_{TST}$ where $\langle . \rangle_{TST}$ represents an average over the set restricted to TST-trajectories.

The standard approach to calculating LDPs is to follow back trajectories using contemporaneous velocity fields (including diabatic heating if relevant) and temperature fields. The novel aspect of the calculations presented in this paper is that the relative sensitivity of temperature variations and transport variations are assessed by generalising the standard approach so that the sampled temperature field is displaced forward or backward in time relative to the original trajectories to arrive at a perturbed estimate of water vapour concentrations.

This approach is depicted schematically in Fig. 1, using an example of three particular trajectories for each of two years. In Fig. 1a temperatures (colours) and winds/trajectories (lines) co-vary from year to year ('standard estimate'). In Fig. 1b the temperature field varies from year to year, but the trajectories are repeated from a given year ('time-shifted-transport estimate'). In Fig. 1c the temperatures are repeated from a given year, but the trajectories vary from year to year ('time-shifted-temperature estimate').

To account for the seasonal cycle in a realistic way, the time-shifted LDP calculations are performed for each month with wind and temperature fields shifted by multiples of one year, so that back trajectories are calculated using time series that start in the same month but from a different year.

To describe these calculations the notation $\text{SMR}_{\text{LDP}}(\boldsymbol{x}_0, t_0^{(\text{traj})}, t_0^{(\text{T})})$ will be used, meaning the SMR evaluated at the LDP along a back trajectory defined by velocity fields starting at $\boldsymbol{x}_0$ at time $t_0^{(\text{traj})}$ and by temperature fields starting at $\boldsymbol{x}_0$ at time $t_0^{(\text{T})}$. This value could be regarded as an estimate for the water vapour concentration at $(\boldsymbol{x}_0, t_0^{(\text{traj})})$ that would be obtained if the temperature field everywhere along the trajectory was displaced in time by $t_0^{(\text{T})} - t_0^{(\text{traj})}$ or an estimate for the concentration at





$(\boldsymbol{x}_0, t_0^{(\mathrm{T})})$ that would be obtained if the trajectory was calculated using velocity fields that were displaced in time by $t_0^{(\mathrm{traj})} - t_0^{(\mathrm{T})}$.

The corresponding tropical estimate is then written as $\overline{\mathrm{SMR}}(t_0^{(\mathrm{traj})}, t_0^{(\mathrm{T})}) = \langle \mathrm{SMR}_{\mathrm{LDP}}(\boldsymbol{x}_0, t_0^{(\mathrm{traj})}, t_0^{(\mathrm{T})}) \rangle_{TST}$.

## 2.2 Reanalysis dataset

The reanalysis dataset used was ERA-Interim (Dee et al., 2011). The trajectories reported in Liu et al. (2010) and Fueglistaler et al. (2013) formed the basis to calculate LDPs in the standard and time-shifted cases. The calculations for ERA-Interim were performed using potential temperature ($\theta$) as a vertical coordinate (with vertical motion relative to $\theta$ surfaces specified

by diabatic heating), with winds, temperatures and diabatic heating rates provided at 6 hourly intervals at $1° \times 1°$ horizontal resolution. The trajectories were initialised from the 83 hPa surface on the first of each month. The period considered was 1999-2009 during which there was significant observed interannual variability in lower stratospheric water vapour.

Liu et al. (2010) have noted that lower stratospheric water vapour estimates based on LDPs calculated using ERA-Interim winds, temperatures and diabatic heating rates are significantly lower than observed. This may be, in part, because TTL tem-

peratures in ERA-Interim are lower than in most other reanalysis datasets and satellite observations (Tegtmeier et al., 2020), and also because of the simplified set of processes assumed by the LDP method. Different approaches have been taken when interpreting LDP estimates using ERA-Interim. Liu et al. (2010) found that the addition of a temperature correction of 3 K reduces the bias in both annual mean concentration and in the amplitude of the seasonal cycle. Other authors, for example Ploeger et al. (2013), found the inclusion of empirical corrections to represent the effects of the microphysical details of parti-

cle formation and sedimentation also act to increase the estimates of water vapour concentrations. In this paper, since our focus is on the relative effect of different processes, rather than any comparison against observed values, we simply accept that the predicted water vapour concentrations have a systematic low bias relative to observations. To illustrate this point, and to provide background for the studies to be reported in the remainder of the paper, Fig. 2 shows the average water vapour concentrations predicted by our LDP calculations over the period 1999-2009 together with the corresponding concentrations taken from the

Stratospheric Water and Ozone Satellite Homogenized (SWOOSH) database (Davis et al., 2016). The difference between the concentrations is typically about 1.5 ppmv, but the pattern and amplitude of seasonal and interannual variation agrees well between SWOOSH and the LDP calculation.

## 2.3 Chemistry-climate model simulation

The chemistry-climate model used in this study is a version of the Met Office Unified Model with United Kingdom Chemistry

and Aerosols (UM-UKCA) based on the Hadley Centre Global Environmental Model 3 (HadGEM3) vn7.3 (Hewitt et al., 2011) with interactive stratospheric chemistry (Morgenstern et al., 2009). The model version was used in Ko et al. (2013). The model is run at N48 resolution ($3.75° \times 2.5°$ longitude-latitude grid) with 60 vertical levels up to an altitude $\sim$84 km. The model solves the three-dimensional equations of motion with vertical velocity as a prognostic variable. The radiative transfer scheme is Edwards and Slingo (1996) updated to use a correlated-k method (Cusack et al., 1999). The stratospheric chemistry

module uses the Fast-Jx photolysis scheme (Telford et al., 2013) and explicitly considers seven chlorine ($CCl_4$, CFC-11, CFC-12, CFC-113, HCFC-22, $CH_3CCl_3$, $CH_3Cl$) and five bromine (H-1211, H-1301, $CH_3Br$, $CH_2Br_2$, $CHBr_3$) source gases.



The model includes a non-orographic gravity wave drag parametrization scheme and simulates a spontaneous QBO (Scaife et al., 2002).

The UM-UKCA simulation analysed in this study is a timeslice simulation with perpetual year 2000 boundary conditions for greenhouse gases, ozone depleting substances, HFCs, aerosols, tropospheric ozone precursor species and orbital parameters. Following Ko et al. (2013), year 2000 sea surface temperature and sea ice boundary conditions are taken as a 10-year average from a coupled ocean-atmosphere HadGEM1 simulation with observed historical external forcings. The model is spun-up for 10 years and a subsequent 49 years of data are used to calculate back trajectories (see below). This model is generally representative of resolved tropical tropopause and stratospheric processes (Gettelman et al., 2010) but has a warm and wet bias as in other versions of the HadGEM3 climate model (Hardiman et al., 2015). Unlike earlier versions of the UM-UKCA model, the version used here has freely varying stratospheric water vapour that interacts with the radiation scheme. Diabatic heating rate fields were not available from the simulation and the back trajectory calculations therefore use three-dimensional velocity fields (rather than two-dimensional velocity fields plus diabatic heating). The model uses hybrid height as a vertical coordinate and some straightforward modifications were required to adapt the OFFLINE code to this. Trajectories were initialised in the middle of each month on the 400 K $\theta$-surface. Other details of the LDP calculation were the same as for ERA-Interim.

## 3 Results – ERA-Interim reanalysis

### 3.1 Interannual variability

This section examines the relative roles of interannual variation in temperatures and transport in determining observed interannual variation of stratospheric water vapour over 1999-2009. In this time, lower stratospheric water vapour decreased markedly in the period 2000-2001 and remained relatively low for several years (Randel et al., 2006; Rosenlof and Reid, 2008). There were several La Niña/El Niño events during the decade, the largest El Niño being in 2002/3 with weaker events in 2004/5 and 2006/7, which may have affected stratospheric water vapour (e.g. Gettelman et al., 2001; Bonazzola and Haynes, 2004; Tao et al., 2019).

Figure 3a shows the standard (contemporaneous temperature and winds) LDP estimated water vapour timeseries (black line) and the alternative time-shifted-transport timeseries (orange lines) for each of the 11 transport years. Figure 3c shows the corresponding deseasonalised timeseries as differences from the average annual cycle of the standard case. Note that, by construction, each of the time-shifted-transport curves in Fig. 3a exactly matches (and is concealed by) the black line during one year. The corresponding time-shifted-temperature LDP estimates are shown as purple lines in Fig. 3b and 3d.

There is a strong contrast between the estimated stratospheric water vapour mixing ratios from the time-shifted-temperature and time-shifted-transport calculations, which is particularly clear in Fig. 3c and 3d. The interannual variation in the time-shifted-transport calculations shows much better agreement with the standard calculation than the time-shifted-temperature calculations. The interannual variation predicted in the standard calculation is typically ±0.5 ppmv. The difference between the time-shifted-transport calculation and the standard calculation is typically less than than 0.2 ppmv and almost always greater than zero. Table 1 shows that the coefficient of determination ($R^2$) between the standard calculation and the time-shifted-



transport calculation is greater than 70 % for all but one (2005) choice of the transport year. The time-shifted-temperature calculations, on the other hand, show little coherence with the standard calculation and the $R^2$ value for the two timeseries is typically 10 % or less.

The experiment design leads to a relationship between the time-shifted-transport and time-shifted-temperature curves in Fig. 3a and 3b that will be briefly explained using year 1999 as an example. In Fig. 3a, the 11 orange lines for 1999 represent
wind conditions for each of the years in combination with the temperature annual cycle for 1999; arranged in sequence these replicate the individual timeseries (purple line) in Fig. 3b with 1999 temperatures repeated. To give an overall measure of the behaviour of the time-shifted cases, the 1999 time-shifted-transport curves are averaged over all choices of transport year except that matching the standard case (1999); this is shown as the 1999 portion of the purple curve in Fig. 4a. The full curve is thus a sequence of annual cycles that depict the mean LDP water vapour over trajectories from alternative years, transported
through the temperature field for the matching calendar year (using notation introduced in Sect. 2.1 this is the average over $t_0^{(traj)}$ for choices of time-shifted $t_0^{(T)}$). The corresponding average for the time-shifted-temperature calculation is shown as the orange curve, which represents the mean LDP water vapour over trajectories for the matching calendar year, transported through temperature fields from all alternative years (average over $t_0^{(T)}$ for choices of time-shifted $t_0^{(traj)}$).

In Fig. 4a, the previously identified poor agreement in the time-shifted-temperature calculation is further demonstrated by
the fact that the orange curve shows similar variations from one year to the next, whereas the purple curve captures much of the structure of the interannual variation in the standard calculation. Figure 4b shows the difference between the average of the time-shifted-transport calculations and the standard calculation, with the climatological annual cycle removed. The fact noted previously that the difference is typically less than 0.2 ppmv is evident, but it can be seen that there are two anomalous years, 1999 and 2008, in which the differences are significantly larger, implying that the combined variation of temperatures and
transport is important in those two years. Both 1999 and 2008 began relatively dry, followed by a transition to relatively wet conditions late in the year. Note that the largest difference between the time-shifted-temperature calculations and the standard calculation in both these years is in the July-October period.

Figure 4b shows some interesting seasonal variations in the differences between the time-shifted-transport and the standard calculations with a pattern that applies across most of the 1999-2009 period. The average difference is typically very small in
the October-January period and tends to be largest in June-July (though the detailed structure of the time variation is different from year to year). The difference between the time-shifted-transport and standard calculations is predominantly positive, suggesting that the combination of contemporaneous transport and temperatures in a given year is efficient at sampling low temperatures relative to generic transport through the same temperature field. But this efficiency does not apply in the October-January period and indeed the difference during that period is negative in some years.

The overall conclusion from the above results is that variation in temperature is the main driver of year-to-year variability in stratospheric water vapour entry, accounting for more than 70 % of the interannual variation. However, in certain years, and on the basis of the limited record analysed here these tend to be a subset of years that are relatively dry, the combination of temperatures and transport together is important for capturing the observed water vapour fluctuations.





As noted previously, there is a noticeable decrease in stratospheric water vapour in late 2000 which has been the focus of
several papers (e.g. Randel et al., 2006; Rosenlof and Reid, 2008; Fueglistaler et al., 2013; Hasebe and Noguchi, 2016), not
least because, looking at longer timescales, it marks a transition between a relatively moist period prior to 2000 and a relatively
dry period after 2000. Hasebe and Noguchi (2016) made a detailed trajectory based study of the late 2000 drop in which they
identified spatial redistribution of Lagrangian dry points for this period relative to the same period in preceding years. They
associated this redistribution with the combined effects of reduction in the strength of the south Asian monsoon circulation and
of changes in the spatial distribution of TTL temperatures. However, without further analysis a given LDP redistribution cannot
be attributed to any particular contribution of changes in temperatures and changes in transport. The results in Fig. 3a, 3c and
4b show that both standard and time-shifted-transport calculations capture the reductions in 2000-2001, with little variation of
the result across the choices of year for time-shifted-transport calculations. That water vapour variation has been demonstrated
to be relatively less sensitive to transport than to temperatures indicates that the changes in TTL temperatures were the primary
driver for the drop in water vapour in late 2000.

## 3.2    Annual cycle

This section considers the role of temperatures and transport in setting the annual cycle in stratospheric water vapour using the
method discussed in Sect. 3.1.

First, some detailed diagnostics from the standard calculation with contemporaneous temperatures and transport are dis-
cussed. These diagnostics are constructed from the same LDP calculations over the period 1999-2009 as in Sect. 3.1, but
focusing on the average properties of the annual cycle rather than interannual variation.

Figure 5 shows characteristic features of the ERA-Interim LDP calculations associated with the average annual cycle in
tropical lower stratospheric water vapour. This expands on the results shown in Fig. 6 of Fueglistaler et al. (2013). Figure 5a
shows water vapour at 83 hPa in SWOOSH observations (black dotted line) and the $SMR_{LDP}$ prediction from back trajectories
initialised at that level (solid black line with coloured diamonds marking each initialisation date). In the SWOOSH data, a
clear annual cycle is seen with a minimum in March/April of around 3 ppmv and maximum in November of 4.8 ppmv, i.e.
a peak-to-peak amplitude of 1.8 ppmv. The $SMR_{LDP}$ calculation shows a very similar phase, with a minimum of 1.7 ppmv
and maximum of 3.1 ppmv, i.e. a peak-to-peak amplitude of 1.4 ppmv. The dry bias and reduced annual cycle amplitude in
$SMR_{LDP}$ relative to observations has previously been noted by Liu et al. (2010).
Figure 5b shows the seasonal variation of $SMR_{LDP}$, with each coloured line corresponding to a coloured diamond in Fig.
5a. Each line shows the average $SMR_{LDP}$ over the subset of back trajectories for which the LDP is encountered in each month
of the set's history. Figure 5b also shows a dotted line which is the average saturation mixing ratio over the 'driest 10 %' of
all vertical profiles within the tropics 30° N – 30° S in a given month. The driest 10 % of profiles is determined based on
the lowest saturation mixing ratio identified in each vertical profile. (Note the mean saturation mixing ratio over all tropical
profiles is around 9.7 ppmv, significantly wetter that the mean of the driest decile.) The dotted line therefore varies roughly in
proportion to the saturation mixing ratio associated with the Eulerian mean temperature in the tropics (30° N – 30° S). This
Eulerian measure has been used in some papers to approximate the saturation mixing ratio in the absence of detailed trajectory





information (see, for example, Garfinkel et al., 2013; Oman et al., 2008). To a large extent the Eulerian and Lagrangian measures show similar seasonal variation, but the efficiency of sampling of low saturation mixing ratios by the trajectories is

significantly better than driest decile, as indicated by the lower mean value of the former by 1 ppmv in boreal winter and up to 2 ppmv in boreal summer and autumn. The coloured lines approximately superimpose, indicating that to a large extent the $SMR_{LDP}$ sampled in a particular month does not strongly depend on when the trajectories are initialised in the year.

The average $SMR_{LDP}$ for a given initialisation month in Fig. 5a is a weighted average of the values from the equivalent coloured line in Fig. 5b. These weightings are the fraction of LDPs occurring in each of the months preceding the initialisation

month, as shown in Fig. 5c. For example, for back trajectories released in August (dark blue line), no LDP events occur in the first month preceding release. The second month preceding release sees around 25 % of LDPs, the third month around 35 %, and the fourth month around 15 % with the fraction reducing systematically further back in time. Taking these weightings into account, it can be seen that the parts of the lines in Fig. 5b that seem to be outliers make only a small contribution to the average $SMR_{LDP}$ in Fig. 5a.

For back trajectories initialised in June-October, the largest number of LDPs are found in the second or third month preceding release. For back trajectories initialised in November-May, the largest number of LDPs are found in the first month preceding release. This is likely to be due to a combination of two effects. The first is that tropical lower stratospheric upwelling in the Brewer Dobson circulation is stronger (Butchart, 2014) and the vertical minimum in temperatures in the TTL is at higher altitudes in boreal winter and spring (e.g., Kim and Son, 2012; Randel et al., 2003), therefore the transit time from the coldest

region to the 83 hPa level is shorter. The second reason is that the seasonal variation in temperatures means that back trajectories released in boreal summer are likely to experience colder temperatures further back in time hence lengthening the distribution of LDPs backward time, and vice versa for those released in boreal winter.

The seasonal distribution of weights explains why the annual cycle in $SMR_{LDP}$ at 83 hPa in Fig. 5a is distorted in shape relative to the dry point annual cycle in Fig. 5b, with a longer moistening time (April-November) and faster drying time

(November-March). The slow increase in spring and summer is due to broader transit time distributions that 'stretch out' this part of the annual cycle signal. Conversely, the narrower distributions in winter give a more immediate signal and hence a relatively rapid decrease in $SMR_{LDP}$.

We now apply the time-shifted trajectory approach previously used in Sect. 3.1 to assess the relative roles of transport and temperatures in determining the annual cycle variation. Here the shift applied is by a given number of months rather than by

a number of years. The specific choices are that the shifted time series in either velocity (hence transport) or temperatures are initialised in the months of February, May, August and November in 2004 (as the non-shifted time series are initialised for each month of 11 years and there are 8 choices of shifted timeseries, the total number of back trajectories being considered is about $6 \times 10^6$ LDPs).

The results for these cases are shown in Fig. 6a and 6b for time-shifted transport and Fig. 6c and 6d for time-shifted tem-

perature calculations. Figures 6a and 6c show results for a three year period 2003-2005 and Fig. 6b and 6d show climatologies averaged over all 11 years. The black curve in each case shows the standard calculation with contemporaneous temperatures and transport. Four curves are shown in each sub-panel corresponding to each month of initialisation of the shifted timeseries.





For example, in Fig. 6a and 6b the dark brown curve shows the time-shifted-transport calculation initialised in February 2004 as indicated on 6a by the matching colour bar. The dark brown curve exactly matches the standard calculation in February 2004

because the shift in the transport initialisation is zero in that month.

Figures 6a and 6b show that with time-shifted-transport the LDP calculation roughly captures the phase of the annual cycle generated by the standard calculation. However, the maximum is significantly overestimated and occurs a month or so early. The largest overestimate of the maximum is for transport initialised in February. The smallest overestimate is for transport initialised in August. Since the orange lines of this figure differ only by time-shifted-transport, and because of the seasonality

in upwelling, we can consider an explanation in the contribution of transport pathways to the seasonal distribution of LDPs shown in Fig. 5c. The boreal summer and autumn weights have a broader distribution, meaning that low saturation mixing ratios experienced several months earlier can have a significant effect on the average $SMR_{LDP}$. Therefore, the influence of the higher saturation mixing ratios in boreal summer is relatively reduced. The weighting functions for boreal winter and spring have a much narrower distribution meaning that it is the recent history of saturation mixing ratios that has the strongest effect

on the average $SMR_{LDP}$. This means the combination of initialising transport using August 2004 velocities with temperatures from boreal autumn gives lower average $SMR_{LDP}$ than other combinations of transport, since the latter tend to sample more recent and therefore higher saturation mixing ratios. The more recent sampling also tends to shift the time of the maximum average LDPs towards the time of the maximum saturation mixing ratios in the TTL.

Turning to Fig. 6c and 6d, the LDP calculation with time-shifted-temperatures cannot reproduce a recognisable seasonal

variation, confirming that the annual cycle in $SMR_{LDP}$ is primarily driven by the annual cycle in temperatures. However, the line corresponding to August 2004 initialisation of temperature does indicate that seasonal variation of transport has some effect. For this temperature initialisation date, the $SMR_{LDP}$ has a marked minimum in July-October, which appears consistent with the behaviour of the time-shifted-transport calculations discussed above. The transport through the TTL of trajectories initialised in the July-October period is relatively slow, therefore they can be influenced by low saturation mixing ratios experienced several

months previously. When transport is fast, the average $SMR_{LDP}$ is primarily determined by recent values, which for temperature initialisation in August will be determined by TTL temperatures in June and July and give relatively large saturation mixing ratios.

The results therefore imply that it is the larger values of saturation mixing ratio in late boreal summer/autumn that are most sensitive to seasonal variations in transport. This sensitivity arises because the slow transport, which is evident in the seasonal

LDP distributions (Fig. 5c), allows a large proportion of pathways to experience relatively low temperatures and hence low saturation mixing ratios in boreal spring, thereby constraining values that already give the maximum in the annual cycle.

### 3.3  Role of sub-seasonal time variability in tropopause temperature

This section considers the effect on stratospheric water vapour of temperature variations on sub-seasonal timescales, to the extent that these variations are resolved in ERA-Interim data.

Figure 7 shows the structure of TTL temperature and its variability in ERA-Interim on sub-daily to seasonal timescales. The separation of the amplitude of variability on different timescales is made by taking running means of the temperature



field at each gridbox using different time windows, and taking the root–mean square difference between the timeseries. Figure 7b shows sub-monthly temperature variability as the difference between 6-hourly data and a 30-day running mean. In the tropics, the sub-monthly temperature fluctuations in ERA-Interim peak along the equator over Africa, the Indian Ocean and the Maritime Continent. The sub-monthly variability can be divided into contributions from different timescales, as shown in Fig. 8. However, we note that the contributions presented in this way share some dependence because the sum of their individual variances underestimates the total variance.

There are many processes that are likely to contribute to TTL temperature variability on sub-monthly timescales. The shortest timescales are likely to be associated with gravity waves generated by convection, with a strong diurnal component (particularly over land) and deep convection (Maritime Continent and West Pacific Ocean) at daily timescales (Johnston et al., 2018); the spatial pattern of regions with higher sub-monthly temperature variability in Fig. 8b appears to be consistent with this. According to analysis of 2 – 20 day filtered variances by Kim et al. (2019), equatorial Kelvin wave and mixed Rossby-Gravity wave activity peaks over east Africa and the east Pacific, respectively. The high sub-monthly variability over the Maritime Continent is also related to 8 – 30 day synoptic variability (Fig. 8c).

Figure 7c shows temperature variability on the subseasonal (30 – 120 day) timescale for which the Madden Julian Oscillation (MJO) is likely to be a significant contributor to variability (Madden and Julian, 1994; Virts and Wallace, 2014). Note that the temperature variability associated with the MJO is not confined to the Indian Ocean/West Pacific regions, that tend to be associated with MJO precipitation anomalies, but is also influenced by the dry Kelvin wave excited by those anomalies. See Virts and Wallace (2014) for further discussion. The local magnitude of subseasonal temperature variability is typically at least a factor of two smaller than the sub-monthly variability. Note that there is significant temperature variability in the subtropics, most likely associated with synoptic systems originating in the midlatitudes and penetrating into the subtropics.

The relative importance of this variability, or indeed any of the other features in Fig. 7, for stratospheric water depends on the effects on $SMR_{LDP}$. To quantify this, the LDP calculations along back trajectories are repeated using temperature time-series filtered by running means with different windows, as described above. Some previous results on the effect of TTL time variability on $SMR_{LDP}$ have been given by Bonazzola and Haynes (2004), by simply time filtering the Lagrangian temperature timeseries, and Liu et al. (2010), who considered the effect of imposed Eulerian temperature variability at different timescales. Here, the approach is to control the variability by time filtering the Eulerian temperature fields that the trajectories sample from.

The results of the LDP calculations with time filtered temperature fields are shown in Fig. 9. As before, the ERA-Interim calculations are for the period 1999-2009. Calculations for UM-UKCA are also shown and will be discussed later in Sect. 4.2. Figure 9a shows the average temperatures at LDPs ($T_{LDP}$) resulting from the trajectories sampling the smoothed temperature fields, with the timescale of averaging on the horizontal axis. Time mean values are shown as grey circles and the average range of the annual cycle in grey vertical lines. The corresponding blue squares and lines show equivalent results for cases where space-time locations of LDPs are not recalculated from the time-filtered temperatures, but the saturation mixing ratios are re-evaluated using the LDPs from the unfiltered data applied to the time-filtered temperature field.



Focusing on the grey circles and lines in Fig. 9a, it is seen that if sub-monthly temperature variability is neglected the average $T_{LDP}$ increases by 2.5 K (i.e. the difference between the $T_{LDP}$ values for 30 days and 6 hours). The corresponding difference from neglecting sub-seasonal temperature variability (i.e. on timescales of less than 60 days or 90 days) is about 3 K. These differences are consistent with that deduced from LDP calculations by Fueglistaler et al. (2013) (see their Fig. 5a).

However, they are much larger than the 0.6 K difference calculated from vertical cold points obtained by removing sub-90 day temperature variability from ERA-Interim temperature profiles, averaging over 5 sites in the West Pacific (7° N, 134 – 171° E across 1990 – 2014; Kim and Alexander, 2015, Fig. 4). Part of this difference may be due to the Kim and Alexander (2015) focus on the equatorial West Pacific which is only one part of the geographical region over which most LDPs are distributed (see Fig. 10). However, it seems likely that an important part of the difference is the sampling effect that is taken into account

by the LDP calculation, but which is ignored by considering vertical profiles at fixed locations. By definition, the LDPs are the location of minimum saturation mixing ratio along the trajectories, which implies that cold points corresponding to LDPs will be cold relative to the overall distribution of vertical cold points. Averaging temperature differences between those implied by instantaneous temperature fields and those implied by seasonal mean temperature fields across LDPs will therefore almost inevitably give larger negative values than averaging those differences for vertical cold points over a season for all locations in

a particular geographical region. A similar effect can be seen in the probability distribution of differences in LDP temperatures arising from a specified addition of random fluctuations to Eulerian temperature fields presented in Liu et al. (2010, Figure 10). The probability distribution for the differences in LDP temperature is displaced towards negative values relative to the corresponding distribution for the Eulerian temperature differences.

Figure 9b shows the corresponding $SMR_{LDP}$ predictions resulting from sampling the different filtered temperature fields

in ERA-Interim. The previously noted 2.5 K increase in LDP temperatures when sub-monthly temperature fluctuations are omitted corresponds to an increase of slightly more than 1.0 ppmv in $SMR_{LDP}$. The corresponding increase from neglecting sub-seasonal variability is around 1.2 ppmv. The relationship between average $SMR_{LDP}$ differences and $T_{LDP}$ differences is therefore about 0.4 ppmv per K, consistent with that implied by the Clausius-Clapeyron relation for temperature 190 K and pressure 90 hPa. Further inspection shows that this holds across the 6 hourly to monthly timescales represented by the results

shown in Fig. 9. Note that Fig. 9 also shows the effect of time filtered temperatures on the magnitude of the annual cycle in $T_{LDP}$ and $SMR_{LDP}$. As the time filtering window is decreased from 30-days to 6-hours, the magnitude of the annual cycle in $T_{LDP}$ is roughly independent of the sampling, while the magnitude of the annual cycle in $SMR_{LDP}$ decreases by about 15 %. Again, this is what is expected from the dependence of saturation mixing ratio on temperature implied by the Clausius-Clapeyron relation. Note, however, that if the sampling interval is increased to 60 days or longer, i.e. excluding sub-seasonal and sub-monthly

variations, the annual cycle magnitudes for both $T_{LDP}$ and $SMR_{LDP}$ decrease.

There seems to be no indication from the results that any particular range of timescales has the strongest impact. Rather, there is simply a systematic decrease of $T_{LDP}$ and $SMR_{LDP}$ as the temperature time sampling interval decreases. The dependence on timescale appears to be roughly logarithmic, i.e. halving any arbitrarily chosen timescale results in an equivalent change in $T_{LDP}$ and $SMR_{LDP}$. This may be a surprise compared to Fig. 8 which shows the magnitude of temperature fluctuations differs





with timescale (0.5 – 1.7 K), and highlights that Eulerian methods cannot be applied directly to estimate stratospheric water vapour.

As previously noted, the time filtering of temperature variability may change the space-time positions of the LDPs. Figure 10 shows the overall effect of time filtering on the distribution of LDP locations, comparing the results from the 6 hourly and the 30 day mean temperature fields. It is clear that removing sub-monthly temperature variability acts to concentrate LDP locations

over narrower regions. The LDPs over the west Pacific, the Maritime Continent and south-east Asia are more geographically confined in the 30-day average case relative to the 6-hour case, and there is an overall ∼ 7 % increase of LDPs in this region (grey dashed box). On the other hand, there is a smaller reduction in the fraction of LDPs occurring over Africa (purple dashed box) with no major change over the tropical Americas (orange dashed box).

The light blue symbols in Fig. 9a and 9b show results where the LDP locations and timing for each trajectory are not re-

evaluated from the original 6 hourly temperature calculation (i.e. the difference between Fig. 10a and 10b is ignored). This approach is potentially attractive because it does not require every trajectory to be searched for a new LDP and was used in Fueglistaler et al. (2013). However, this will inevitably give a larger average $SMR_{LDP}$ than if LDP locations are re-evaluated based on the new temperature field. From Fig. 9 it can be seen that the impact of using fixed LDPs for the ERA-Interim sub-monthly temperature fields is a further increase of 0.8 K in $T_{LDP}$. The total $T_{LDP}$ difference between the standard 6-hourly ERA-

Interim temperatures and the case with sub-monthly variations removed and without recalculating LDP locations is therefore 3.3 K, which is highly consistent with the 3.2 K reported by Fueglistaler et al. (2013). The corresponding effect on $SMR_{LDP}$ is 0.7 ppmv (a 60 % increase from the full 6-hourly calculation). Therefore, even though the overall distribution of LDPs is changed relatively little by the re-evaluation (Fig. 10) the difference is quite substantial and, for this type of calculation, where an alternative temperature (or saturation mixing ratio) field is provided, it is clearly desirable that the effect of re-evaluating

locations of LDPs is taken into account.

In addition to considering the role of time variability in temperature, we now also consider the role of zonal variation. This is motivated in part by the previous use of zonal mean temperature at a fixed pressure level in the TTL as a proxy for the coldpoint in the absence of Lagrangian information (e.g. Hardiman et al., 2015) but also because zonal variation of temperature is now regularly invoked in explanations of dehydration. Fig. 9a and b also show average $T_{LDP}$ and $SMR_{LDP}$ values obtained from

zonally averaged temperatures for the 6-hourly and 30-day mean cases. For 30-day mean temperatures, the effect of zonally averaging temperatures is to increase $SMR_{LDP}$ by about 0.9 ppmv. This case is also about 1.8 ppmv higher than the 6-hourly temperature case with no zonal averaging, indicating that sub-monthly time variability and longitudinal variation account for about equal proportions of the dehydration relative to a zonal-mean monthly average picture.

## 4    Results - Chemistry-climate model

This section presents results from the UM-UKCA chemistry-climate model described in Sect. 2.3 for comparison with the ERA-Interim results in Sect. 3. Given the absence of interannual changes in boundary conditions in the model simulation (see Sect. 2.3), emphasis here is placed on the seasonal variation and on the role of sub-seasonal and sub-monthly variability.



### 4.1 Seasonal and interannual variation

The characteristics of the modelled LDPs over the climatological annual cycle are shown in Fig. 11, which can be compared
with those for ERA-Interim in Fig. 5. As in Fig. 5, Fig. 11a shows water vapour in the tropical lower stratosphere, Fig. 11b
shows Eulerian and Lagrangian estimates of final dehydration saturation mixing ratios, and Fig. 11c shows the relative fraction
of LDPs in each month as a function of initialisation date. Modelled tropical mean Eulerian mean water vapour in the lower
stratosphere at 80 hPa is shown for comparison in Fig. 11a.

As noted in Sect. 2.3, the modelled TTL temperatures are significantly warmer than in ERA-Interim (e.g. 11 year mean
tropical and zonal mean temperature at 96 hPa in ERA-Interim is 194.6 K, whereas in the model at 100 hPa it is 199.6 K,
implying that saturation mixing ratios will be higher. This difference extends to the lowest decile of vertical dry point saturation
mixing ratios shown in Fig. 11b, where the model values are 5 – 11 ppmv compared to the reanalysis range of 2.5 – 6.5 ppmv.
Correspondingly, the $SMR_{LDP}$ estimate is wetter throughout the year by about 3 ppmv, with an annual cycle minimum and
maximum of about 4.5 ppmv and 6.4 ppmv, respectively (compared to 1.5 ppmv and 3 ppmv for ERA-Interim). The minimum
in $SMR_{LDP}$ over the annual cycle is also later by about a month in the model.

While care must be taken in comparing the reanalysis trajectories released from 83 hPa with the model trajectories released
from the $\theta = 400$ K level, there do appear to be differences in the sampling of temperatures by the trajectories in the reanalysis
and model. The fraction of LDPs per month shown in Fig. 11c are lower for the model, reaching no higher than 35 % whereas
for the reanalysis the maximum values are typically 50 %. Furthermore, the maximum contribution typically comes from the
second to fourth month prior to release (i.e. further into the back trajectory history) rather than from the first or second month.
Thus, the $SMR_{LDP}$ for a given month in the model is determined by LDPs distributed over a broader range of times than in the
reanalysis. These results suggest that the trajectory transit times from the LDP to the 400 K level in the model are typically
longer than from the LDP to the 83 hPa level in ERA-Interim. This may be partly due to differences in the distance and speed
of vertical transport. The average pressure of LDPs is higher in the model (110 hPa) than in ERA-Interim (94 hPa), indicating
a lower altitude and therefore further vertical distance from the initialisation level. Secondly, vertical advection in the model
may be weaker than in ERA-Interim (since relatively warmer TTL temperatures lead to weaker diabatic heating).

Another difference between model and reanalysis may be seen from the coloured lines in Fig. 11b. The tails of the saturation
mixing ratios for trajectories initialised in boreal autumn and winter show higher values than other curves for the same month,
implying that those back trajectories for which the LDP lies in the first month or two contribute anomalously wetter LDPs
than the average over all LDPs occurring during that period. This behaviour is particularly clear for trajectories released in the
September-February period and is not seen in the reanalysis calculation (Fig. 5b). These trajectories must therefore not sample
the coldest regions of the TTL efficiently on their path to the troposphere. The model trajectories use vertical velocities rather
than diabatic heating rates and these particular trajectories would have experienced large vertical velocities, perhaps associated
with the model representation of vigorous tropical convection that penetrates the TTL. However, the difference in saturation
mixing ratios between model and reanalysis for these trajectories that rapidly transit from the troposphere are quite difficult
to explain – note that the proportion of back trajectories that reach the troposphere within one or two months is higher for the





reanalysis calculation but, in contrast to the model, those do not seem to sample unusually large saturation mixing ratios at their LDPs. Having noted that uncertainty, it is also important to emphasise that these trajectories are of limited significance for the overall seasonal variation of water vapour concentrations, since they make only a limited contribution, at most 30 %
(Fig. 11c) to the monthly mean concentration in the lower stratosphere.

We now consider the roles of transport and temperature variations in setting the average annual cycle in stratospheric water vapour in the model. Figure 12 shows the effect of time-shifted-temperature and time-shifted-transport on the $SMR_{LDP}$ annual cycle for the model simulation, this can be compared with Fig. 6 in Sect. 3.2 for ERA-Interim. The left-hand panels show three specific years from the model simulation and the right-hand panels show the average annual cycle generated from 11
years. As before, the black curve shows the results from the standard $SMR_{LDP}$ calculation with transport and temperature co-varying but the time-shifting is chosen so that temperature or transport time series are initialised either in February or in August of year 8 of the simulation, corresponding respectively to the drying and moistening phases of the $SMR_{LDP}$ seasonal cycle. For the time-shifted-transport calculation (upper panel) the overall pattern of $SMR_{LDP}$ over the annual cycle is reproduced, with the minimum value very close to that in the standard calculation, but the maximum value overestimated by about 30 %.
However, for the time-shifted-temperature calculation the pattern of $SMR_{LDP}$ variation is completely lost. For initialisation using February temperatures, the predicted seasonal variation is very weak, whereas for initialisation using August temperatures there is some seasonal variation, with smaller mixing ratios in August and larger mixing ratios in March. These results are broadly similar to those obtained for ERA-Interim (Fig. 6), with some differences in the details. For example, in ERA-Interim initialisation using fixed transport tends to give a maximum saturation mixing ratio that is a month or two earlier than in the
standard calculation, but this difference is reduced in the climate model calculation. This may be because the seasonal variation in the distribution of LDPs, as indicated by the lower panels of Fig. 5 and 11, is stronger for ERA-Interim than it is for the model. This would also explain the smaller amplitude in the model relative to ERA-Interim of the seasonal variation predicted by fixed initialisation of temperatures in August, recalling from Sect. 3.2 that in ERA-Interim this variation arises because trajectories initialised in the June-November period sample temperatures over a broader range of times than those in other
months. This characteristic is present to some extent in the model calculations, but is not as strong as in ERA-Interim.

As noted previously the model simulation was carried out with fixed boundary conditions and greenhouse gases. There is therefore no external forcing of interannual variability by, for example, sea surface temperature variations. However, there is some internally generated interannual variability, dominated in the tropics by the model's QBO and there is corresponding interannual variability in stratospheric water vapour. The time-shifting approach may be used, as previously, to study the
relative importance of temperatures or transport in this variability. Figure 13 shows results of time-shifted-transport and time-shift-temperature calculations, with $R^2$ values reported in Table 1. These results indicate that in the climate model, as in ERA-Interim, it is variability in TTL temperatures that is responsible for the largest part of the variability in water vapour entering the stratosphere. As was found for ERA-Interim, the effect of time-shifted-transport almost invariably overpredicts the simulated water vapour values, but this overestimate is very small for the seasonal minimum in each year.



## 4.2 Sensitivity to sub-seasonal time variability in tropopause temperature

The examination of the effect of subseasonal variations in temperatures on dehydration, reported in Section 3.3 for ERA-Interim, is now repeated for the climate model. As before the approach is to construct Eulerian temperature fields smoothed on different timescales and to recalculate the LDPs using the same trajectories as in the standard calculation. The saturation mixing ratios evaluated at the LDPs are likely to change, both simply as a result of the change in the temperature fields and also as a result of the change in the space-time LDP location.

Figure 9a shows in the black and dark blue points comparable results for $T_{LDP}$ in the climate model to those previously discussed in Sect. 3.3 for ERA-Interim. Note that some positions on the horizontal axis differ from those of ERA-Interim as not all of the temporal filter window sizes match those applied in the ERA-Interim case. Since the climate model exhibits a warm bias compared to reanalysis, all the points representing the climate model in Fig. 9a (and correspondingly for the saturation mixing ratio results in Fig. 9b) are systematically displaced relative to those for ERA-Interim. However, our focus here is the sensitivity to time-filtered temperatures rather than the absolute values. The difference in average $T_{LDP}$ between the 30-day mean and 6-hourly temperatures is around 1.7 K, compared to about 2.2 K for ERA-Interim. By this measure, temperature variability on sub-monthly timescales is therefore around 20 % lower in the climate model than in ERA-Interim.

Figure 9b, shows corresponding results for $SMR_{LDP}$. The systematic increase in $T_{LDP}$ as lower-frequency temperature variability is removed leads to an increase in $SMR_{LDP}$, as was seen in ERA-Interim. Despite the smaller effect on $T_{LDP}$, the removal of sub-monthly variability leads to a larger increase in $SMR_{LDP}$ of 1.7 ppmv in the model as compared to an increase of 1 ppmv in ERA-Interim. The larger change in $SMR_{LDP}$ in the climate model versus ERA-Interim accompanying smaller difference in temperatures is due to the non-linearity of the Clausius-Clapeyron relation coupled with the warm bias in the climate model. At higher temperatures, the saturation mixing ratio has a larger temperature sensitivity. The sensitivity seen in the results, about 1.0 ppmv per K in the climate model, more than twice that of ERA-Interim, is consistent with a rough estimate from the Clausius-Clapeyron relation, based on a mean $T_{LDP}$ of 193 K for the climate model and 187 K for ERA-Interim.

Also shown in Fig. 9a and b, are the results for the climate model for LDP temperatures and saturation mixing ratios obtained without re-calculating the space-time positions of LDPs. The associated difference in $T_{LDP}$ is about 1 K and in $SMR_{LDP}$ about 1 ppmv, respectively, somewhat smaller and somewhat larger than the corresponding results for ERA-Interim. Interestingly, using zonal mean temperature to calculate LDPs results in an increase in $T_{LDP}$ of around 3 K, which is smaller than the equivalent increase in ERA-Interim of around 4 K. This suggests that the zonal variation in temperature in the model is smaller than in ERA-Interim. However, the corresponding increase in $SMR_{LDP}$ in the model is around 3 ppmv, considerably larger than the increase in ERA-Interim of 1.8 ppmv. These differences can also be explained by the non-linearity of the Clausius-Clapeyron relation. These results show that mean biases in TTL temperature affect both mean stratospheric water vapour and its variability by creating unrealistic sensitivity to temperature variations across timescales.





## 5  Discussion and summary

Our aim in this paper has been to determine the separate effects of temperatures and transport in determining variations in tropical lower stratospheric water vapour. The motivation is partly to identify to what extent the details of temperature and transport variations need to be represented in models, but also that insight into the separate effects of temperatures and transport

is relevant for understanding observed variations in stratospheric water vapour, including the extent to which processes such as convective injection or particle formation and sedimentation may be important.

The technique used in the paper is based on the LDP method, but using different combinations of temperature fields and transport fields than in the standard approach where both fields vary contemporaneously. We have used time-shifted temperature and time-shifted transport to examine separately the contributions of interannual variations in temperatures and transport to

interannual variations in water vapour and also to consider seasonal variation. Extending some of the results of Fueglistaler et al. (2013), we have used LDP calculations based on time-filtered temperature fields to quantify the role of temperature variability on different timescales in setting concentrations of water vapour in the tropical lower stratosphere.

Regarding interannual variability, our results using ERA-Interim show that interannual variability of temperatures alone reproduces 67 – 88 % of the total interannual variability of the LDP estimated water vapour over the period 1999-2009.

Conversely, interannual variability of transport alone reproduces very little (∼ 10 %) of the total interannual variability. A similar conclusion is reached for a chemistry-climate model (UM-UKCA), albeit using a timeslice simulation where drivers of interannual variability are limited compared to observations.

The differences in saturation mixing ratio between the time-shifted-transport and standard calculations are mainly positive, indicating that the time-shifted-transport is systematically less efficient at sampling cold temperatures. In most years, the

largest differences occur in boreal summer and early autumn. In the period 1999-2009, three years (1999, 2007 and 2008) show unusually large differences in saturation mixing ratios between the standard and time-shifted-transport calculations, with differences of up to 0.4 – 0.6 ppmv in boreal late summer and early autumn coincident with the seasonal cycle maximum. In these years the combination of temperature and transport variations appears to be particularly important for capturing the observed variation in stratospheric water vapour. As noted previously, the year 2000 exhibited an unusual drop in stratospheric

water vapour. Our results show that accounting for variations in temperatures alone is sufficient to reproduce this drop in the standard LDP calculation with co-varying fields to within 0.1 ppmv. This contrasts with the results of Hasebe and Noguchi (2016) that the combination of both temperatures and transport was responsible for the 2000 water vapour drop.

With respect to the annual cycle, the time-shifted approach has shown that the seasonal variation of temperatures rather than transport sets the overall pattern of LDP saturation mixing ratio. The time-shifted-temperature calculations completely miss the

overall structure of the annual cycle, but reveal a role for the relatively weaker lower stratospheric upwelling in boreal summer, which means the saturation mixing ratios in late boreal summer are determined by LDPs distributed over several previous months extending back to spring; this results in smaller saturation mixing ratios than if the LDPs were confined to summer alone. The seasonal variation of transport therefore plays an important role in reducing the maximum in the annual cycle of saturation mixing ratios relative to what can be inferred from the seasonal variation and amplitude in Eulerian measures of cold



point temperatures alone. This reduction is counter to the emphasis in some previous papers (e.g. Bannister et al., 2004; James et al., 2008), which have argued that summer-time transport has a moistening effect by allowing pathways from troposphere to stratosphere that avoid the regions with lowest saturation mixing ratios. The generally stronger role for transport in determining water vapour concentrations in boreal summer and autumn is consistent with the seasonal variation noted previously of the amount of interannual variability in water vapour that can be captured by the time-shifted-transport calculation.

Decoupling the time variation of temperatures and transport in this way is an artificial approach and could be justified as realistic only if temperatures and transport were truly independent, which they are not. One important basic dependence is between temperatures and diabatic upwelling rates, particularly on seasonal and longer time scales. It is difficult to find a clear argument that the dependencies between temperatures and transport will inevitably lead to more efficient sampling of cold temperatures than would be the case if the two were independent, not least because it is the time history of trajectories over

periods of months that is relevant. Nonetheless, the time-shifted-transport calculations, for both the reanalysis and model on interannual and seasonal timescales indicate an apparently robust result that time-shifted-transport tends to give less efficient sampling of cold temperatures and consequently higher saturation mixing ratio estimates.

Time filtering was used to quantify the impact of sub-seasonal (30 – 120 days) and sub-monthly (6 hours–30 days) variability in TTL temperatures on stratospheric water vapour concentrations. When sub-monthly fluctuations are included, ERA-Interim

estimates of LDP averages are 2.5 K colder and 1 ppmv drier. The corresponding differences in the model estimates are 1.7 K cooler and drier by 1.7 ppmv. Differences for ERA-Interim when sub-seasonal timescales are included are 0.6 K cooler and 0.3 ppmv drier. The results for both the reanalysis and model indicate that there is no particular range of timescales that has a dominant effect – it is simply that LDP temperatures, and consequently mixing ratios, decrease systematically as shorter timescale variations in temperature are resolved. However it should be remembered that the significance of very short

timescale variations will in reality be limited by the fact that, at such time scales, the simplifying assumption that water vapour instantaneously relaxes to the local saturation mixing ratio becomes less relevant.

The different quantitative relationship between LDP temperature differences and LDP saturation mixing ratio fluctuations of the climate model relative to ERA-Interim can be explained by the non-linearity of the Clausius-Clapeyron relation and the model warm bias. Note if the smaller effect of sub-monthly timescales on LDP temperatures in the climate model versus

ERA-Interim is interpreted as an underestimate of dynamical variability, then the associated bias in LDP mixing ratios is about 0.8 ppmv or around 25 % of the moist bias in the climate model relative to observations. The implication of this is that, in addition to mean temperatures, the representation of sub-seasonal and sub-monthly time variability of TTL temperatures is important for a model to simulate realistic stratospheric water vapour concentrations under current conditions. Kim et al. (2013) noted that most models of the Coupled Model Intercomparison Project Phase 5 (CMIP5) fail to simulate realistic intraseasonal

temperature variability, so this will contribute to the significant mean biases in stratospheric water vapour in climate models.

Kim and Alexander (2015) also demonstrated a qualitatively similar effect of temperature variability on tropical cold-point temperatures and hence on dehydration. However, Kim and Alexander (2015) considered the effect of temperature fluctuations using vertical profiles at selected geographical locations. They concluded that fluctuations on timescales between 1 – 90 days reduced average tropical cold point temperatures in ERA-Interim profiles by about 0.6 K. This is a factor 4 smaller than the





reduction in LDP temperature deduced here. It seems likely that an important part of the difference is the sampling effect, which is taken into account by the LDP calculation but ignored when considering temperature profiles at fixed locations. The difference between the Eulerian estimate of Kim and Alexander (2015) and the LDP estimate presented here points to the limitations of using Eulerian measures of temperature or saturation mixing ratio to explain changes in stratospheric water vapour concentrations (for further discussion, see and Sect. 3.2 and 4.2 Smith, 2020).

In summary, the results presented here have provided clear attribution of the roles of temperature and transport variations in controlling water vapour entry to the stratosphere. Additionally, the results highlight the importance of both mean TTL temperatures and temperature variability across timescales for modelling realistic stratospheric water vapour concentrations.

*Code and data availability.* ERA-Interim is publicly available from https://apps.ecmwf.int/datasets/. The program OFFLINE may be re-
quested from the corresponding author (jws52@cam.ac.uk). The datasets of UM-UKCA simulation variables, UM-UKCA trajectory calcu-
lations and LDP calculations with ERA-I and UM-UKCA and code to produce figures will be made available at an accessible archive.

*Author contributions.* JWS and PHH designed the methodology and developed the OFFLINE program to conduct the experiments. ACM conducted the UM-UKCA simulation. JWS conducted the experiments, analysis and visualisation with supervision by all co-authors. JWS and PHH wrote the initial draft, with contributions from all co-authors.

*Competing interests.* The authors declare that they have no conflict of interest.

*Acknowledgements.* JWS was funded by a Natural Environment Research Council (NERC) Industrial CASE PhD studentship with the Met Office (NE/M009920/1). PHH acknowledges support from the IDEX Chaires d'Attractivité programme of l'Université Fédérale de Toulouse, Midi-Pyrénées. ACM was funded by a NERC Independent Research Fellowship (NE/M018199/1). NB was supported by the Met Office Hadley Centre Programme funded by BEIS and Defra. Programs and data prepared by Stephan Fueglistaler and Sue Liu were a valuable resource for this research.





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



**Figure 1.** Schematic of the method for back trajectory calculations. In this example, there is one spatial dimension and three initialisation dates ($t_0$) each year for two years. The white and black lines refer to back trajectory transport pathways for the two years (calculated from wind fields) and the colours refer to temperatures. Green dots show the initialisation locations ($\boldsymbol{x}_0, t_0^{(\mathrm{traj})}, t_0^{(\mathrm{T})}$), white dots refer to the Lagrangian cold point, and grey dots refer to the limit of trajectory history after 1 year. (a) Standard estimate of sampling co-varying winds and temperature, $t_0^{(\mathrm{traj})} = t_0^{(\mathrm{T})}$; (b) Time-shifted-transport where $t_0^{(\mathrm{traj})} = $ year 3; (c) Time-shifted-temperature where $t_0^{(\mathrm{T})} = $ year 3.





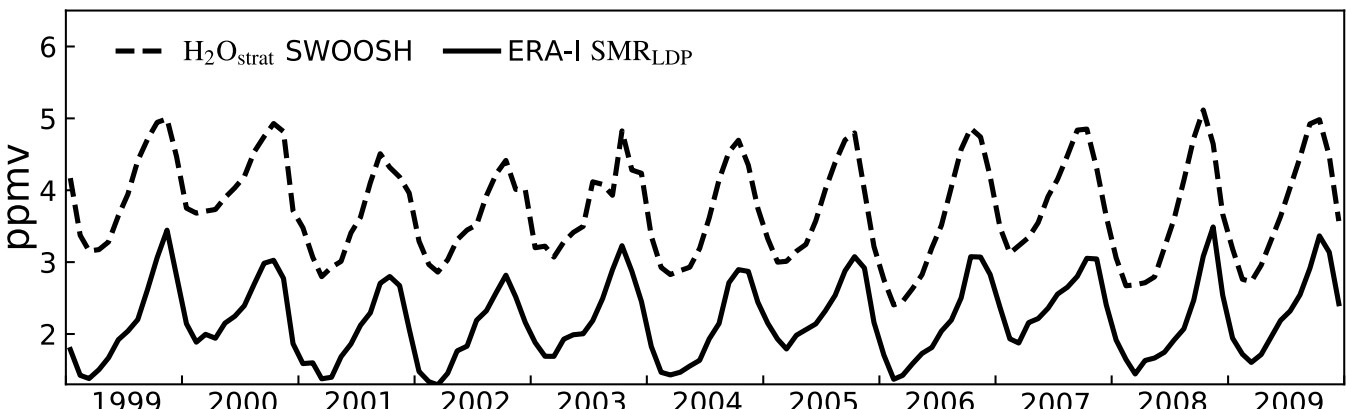

**Figure 2.** Timeseries over 1999-2009 of monthly and zonal mean of water vapour in the tropics (30° N – 30° S) at 83 hPa in SWOOSH (Davis et al., 2016) and Lagrangian Dry Point (LDP) estimates of water vapour entering the stratosphere based on ERA-Interim diabatic back trajectories.

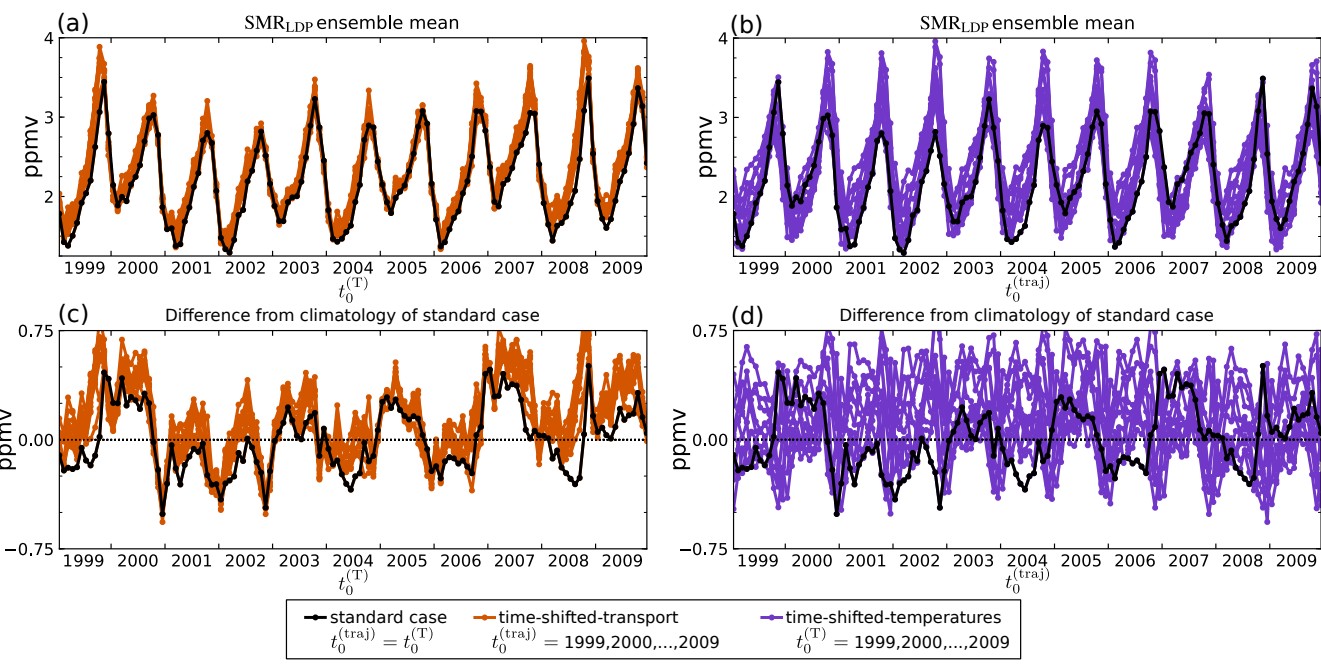

**Figure 3.** Timeseries of $SMR_{LDP}$ for the standard case (black), calculations with time-shifted-transport (orange) and with time-shifted-temperatures (purple). (a,b) Absolute concentrations. (c,d) Anomaly from mean annual cycle of standard case.



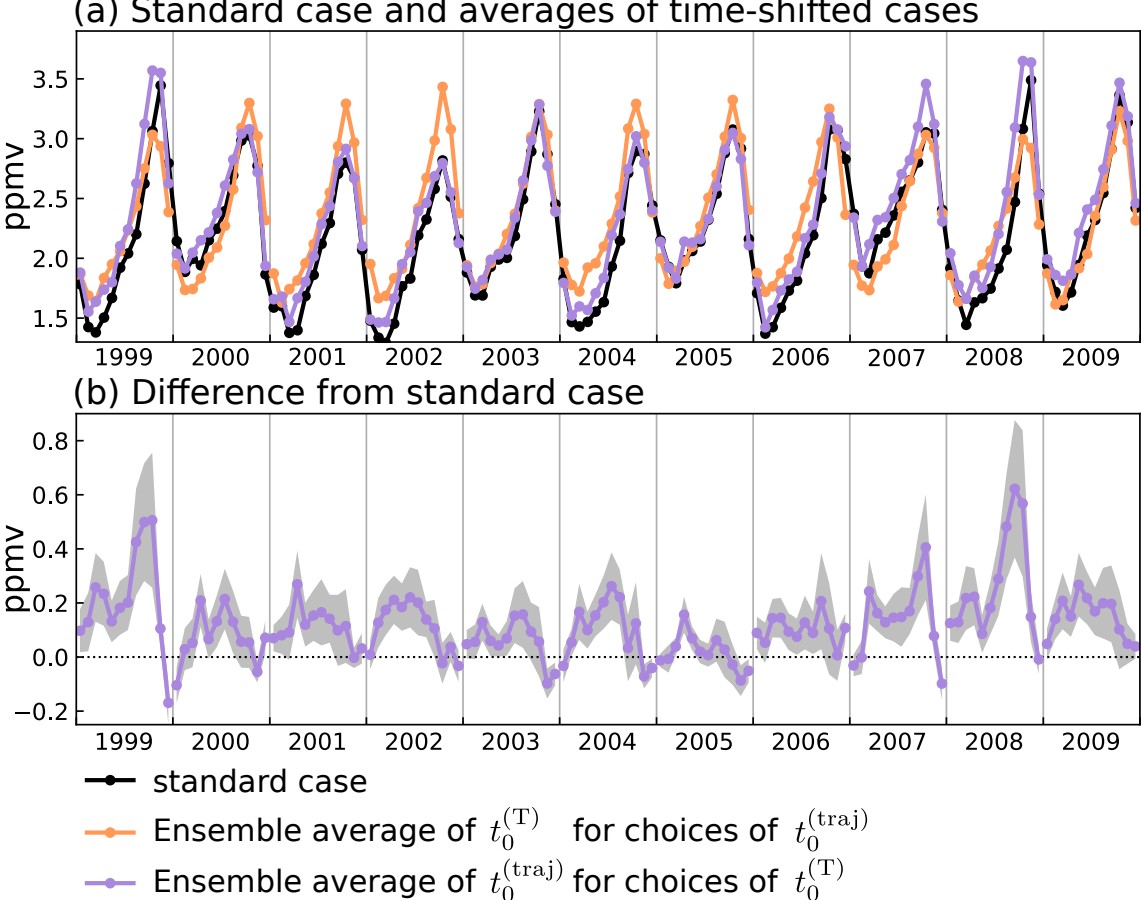

**Figure 4.** (a) Timeseries of $SMR_{LDP}$ for standard case and the in-year average for alternative years of (orange) time-shifted-temperatures and (purple) time-shifted-transport, positioned according to the year being repeated. Further described in Sect. 3.1. (b) Difference between time-shifted-transport cases and the standard case.



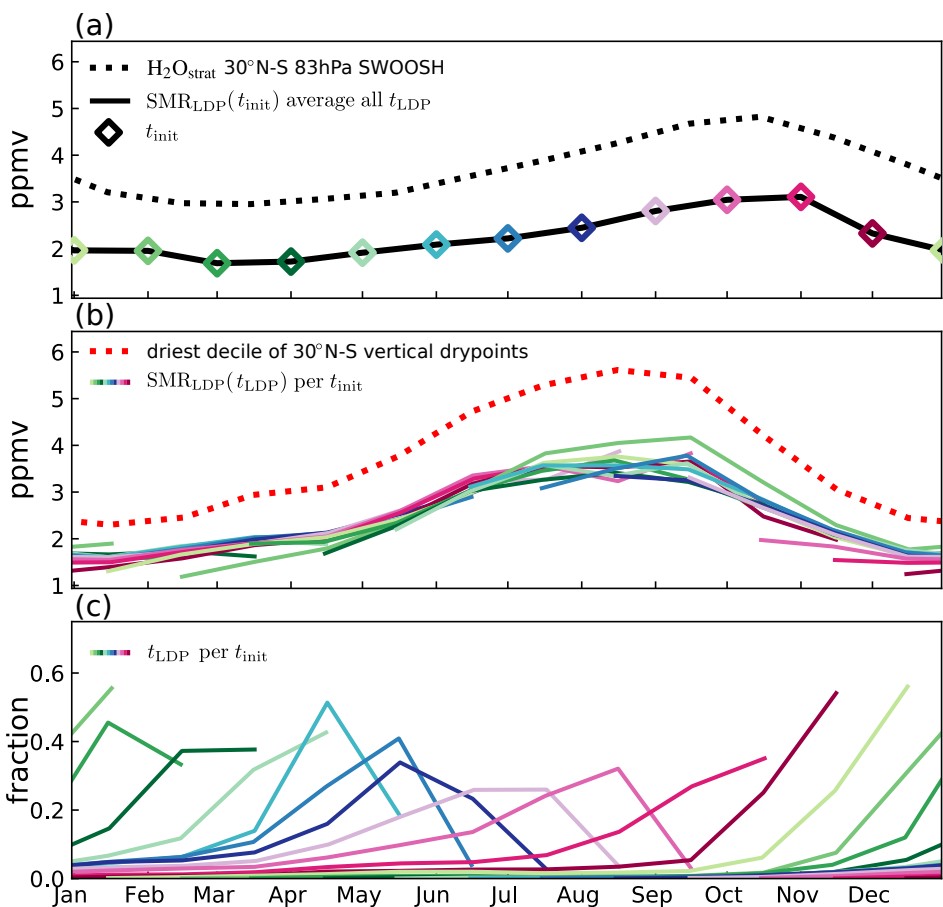

**Figure 5.** (a) Zonal and tropical (30° N – 30° S) mean annual cycle of water vapour at 83 hPa from SWOOSH observations (dotted) and ERA-Interim LDP estimates (black line) with coloured diamonds denoting initialisation dates. (b) Estimates of water vapour concentration at final dehydration location, saturation mixing ratio vertical minimum (vertical dry point, dotted line) and LDPs (coloured line for each release date, $t_{init}$). (c) Fractional contribution of LDP events to $SMR_{LDP}$ estimate in each month for each initialisation date ($t_{init}$) (colours as in (a)). Akin to Figure 6 of Fueglistaler et al. (2013) but showing predicted concentrations from trajectories released within 30° N – 30° S monthly, as well as observations and additional Eulerian estimates of water vapour. See Sect. 3.2 for details.

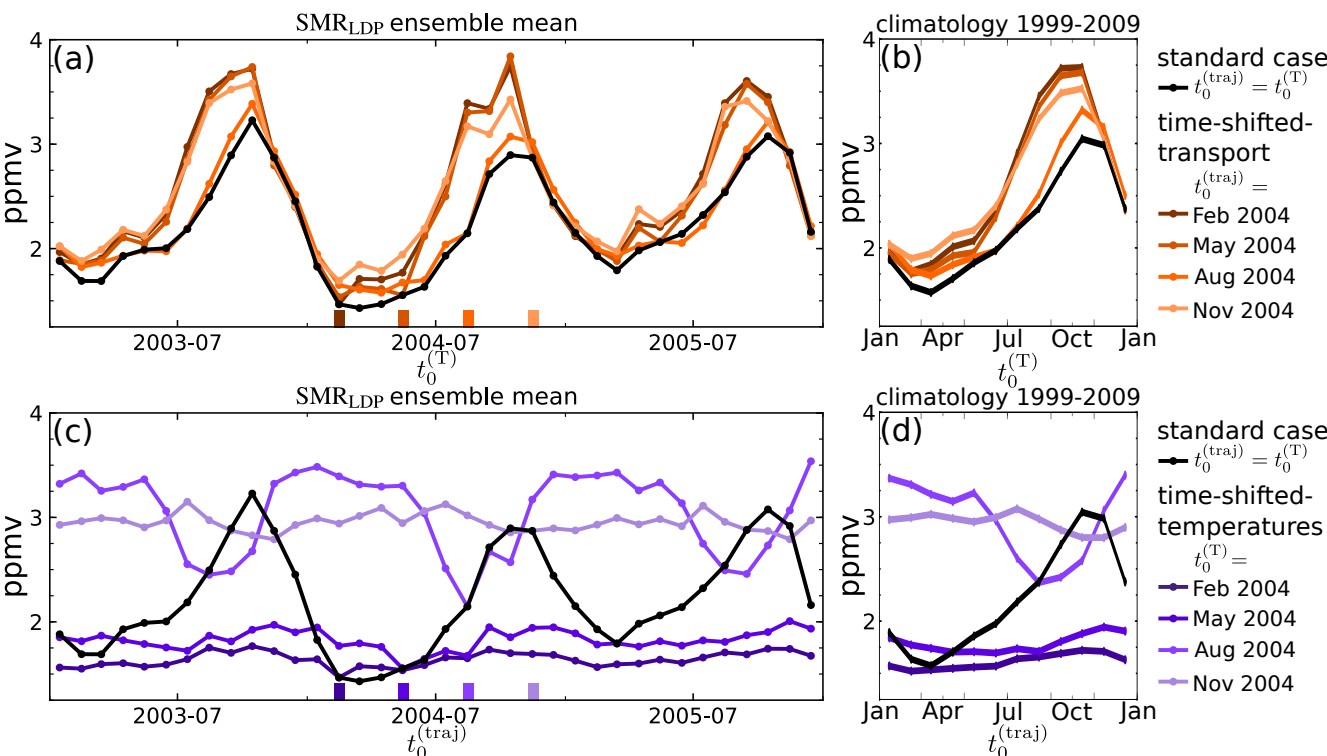

**Figure 6.** Mean SMR$_{LDP}$ between 2003-2005 for ERA-Interim standard co-varying case (black line), (a) wind timeseries initialised to a fixed month of 2004 (orange lines), (b, d) equivalent climatological annual cycles over 1999-2009. (c) temperature timeseries initialised to a fixed month of 2004 (purple lines). Line shade corresponds to particular months marked with ticks, see also legend.

**Figure 7.** 1999-2009 ERA-Interim temperatures at 96 hPa. (a) Time mean, (b) Root-mean-square difference between 6 hourly and 30 day rolling means, and (c) Root-mean-square difference between 30 day and 120 day rolling means.



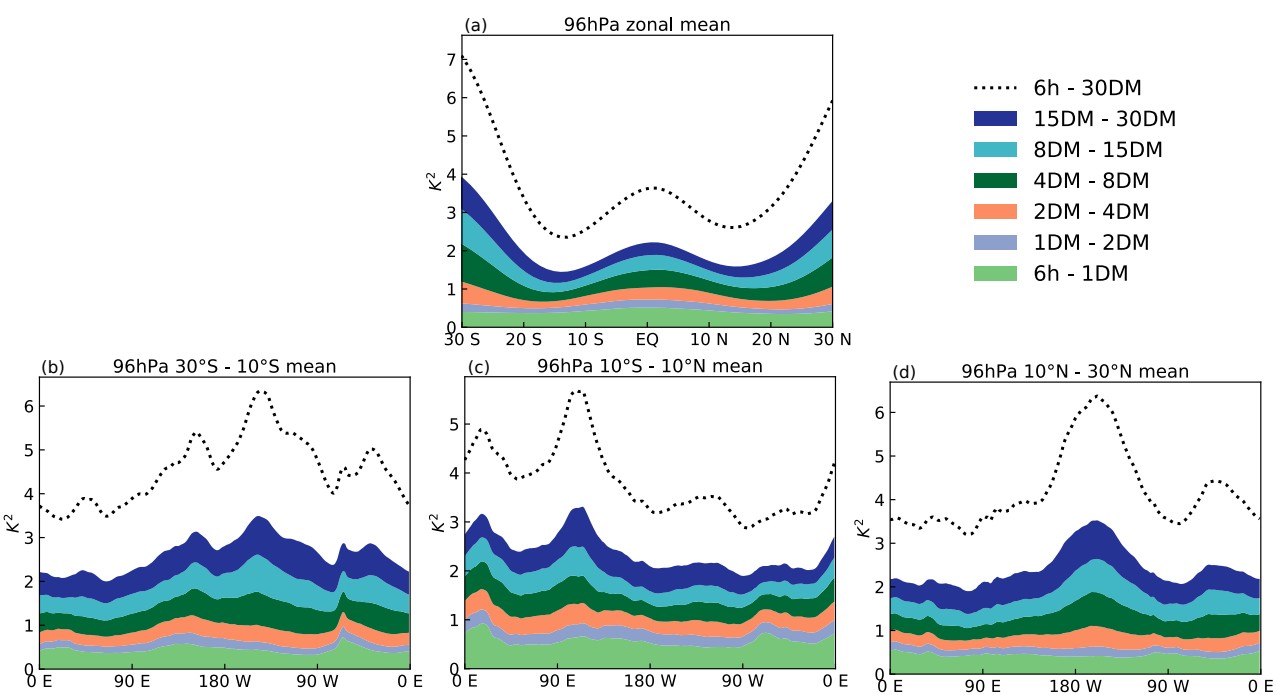

**Figure 8.** Mean square differences between time filtered ERA-Interim temperatures. (a) zonal mean 96 hPa timeseries at each latitude, and as a function of longitude between (b) 30° S – 10° S, (c) 10° S – 10° N, (d) 10° N – 30° N. Mean square difference between 6 hourly and 30 day running mean shown in dotted line, and sub-divisions of sub-monthly timescales are stacked.



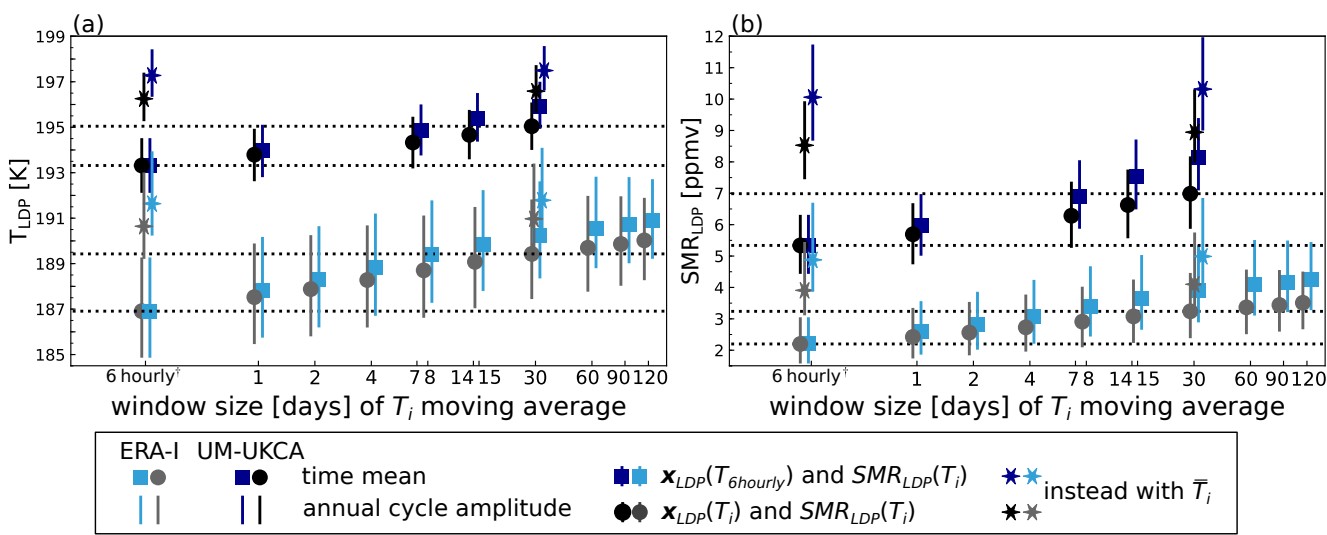

**Figure 9.** Lagrangian dry point calculations for (a) temperature ($T_{LDP}$) and (b) water vapour ($SMR_{LDP}$) experiencing different moving-window time averages in ERA-Interim across 11 years (grey and light blue) and the climate model across 11 years (black and dark blue). Showing time mean (points) and mean amplitude of annual cycle (whiskers). Cases of zonal mean temperature shown as stars. Timescales averaged over include 1-day (1DM) and 120-day (120DM). (†) Note that 6-hourly temperature field is instantaneous, not averaged. Calculations are divided into those which re-evaluate dry point locations according to averaged temperature field (black and grey datapoints) and those which fix dry point locations from 6 hourly calculation (blue datapoints).

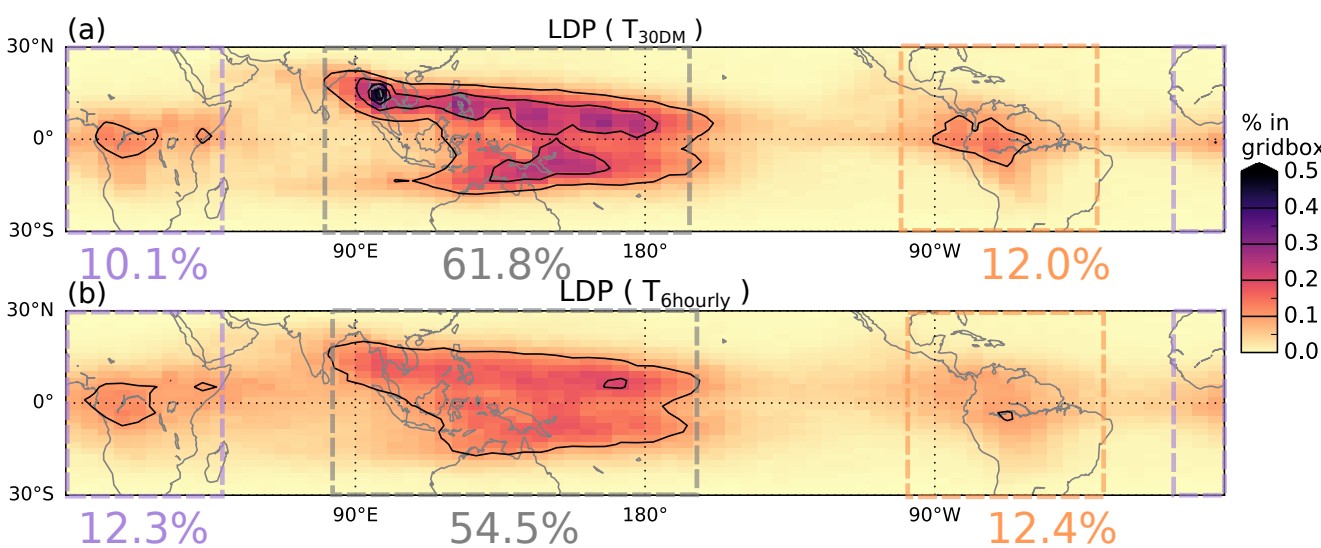

**Figure 10.** Spatial distribution of Lagrangian dry point events for trajectories subject to (a) 30 day mean temperature field and (b) 6 hourly temperatures. The colours show the percent of all LDP events within $4.5° \times 2°$ gridboxes, with black contours plotted every 0.1 %. The numbers give the cumulative percentages within the coloured boxes.



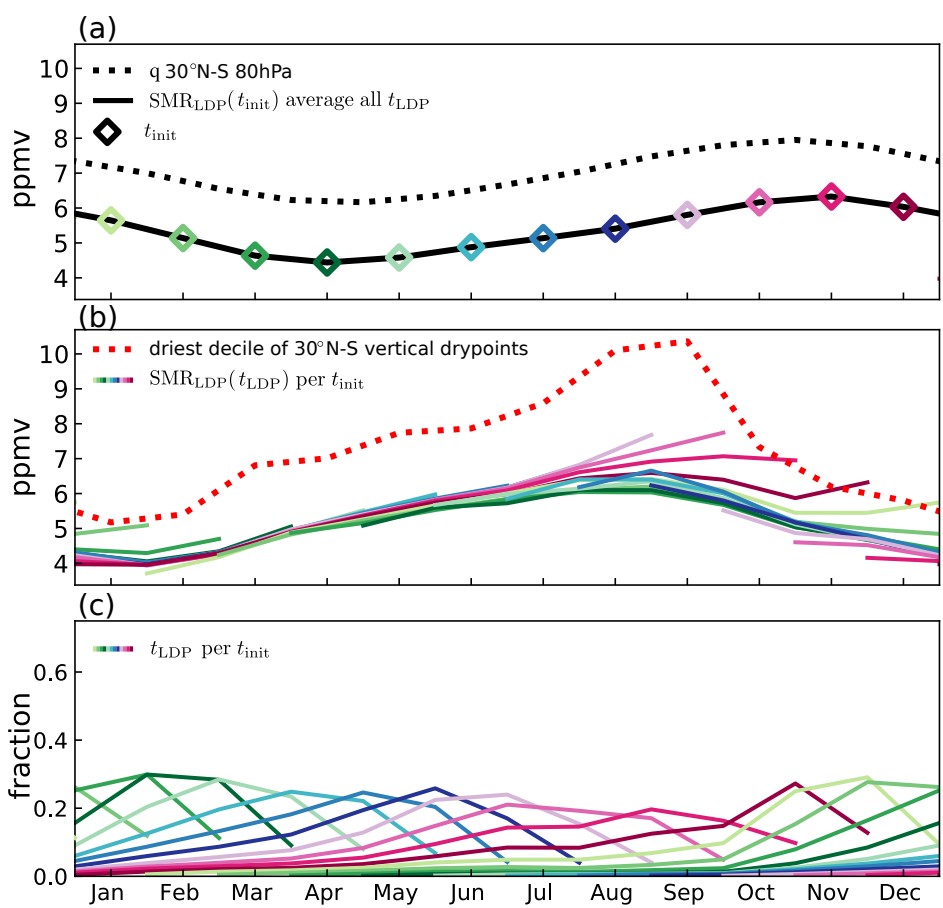

**Figure 11.** Summary results of LDP method as in Fig. 5 but for 11 year average of the climate model.



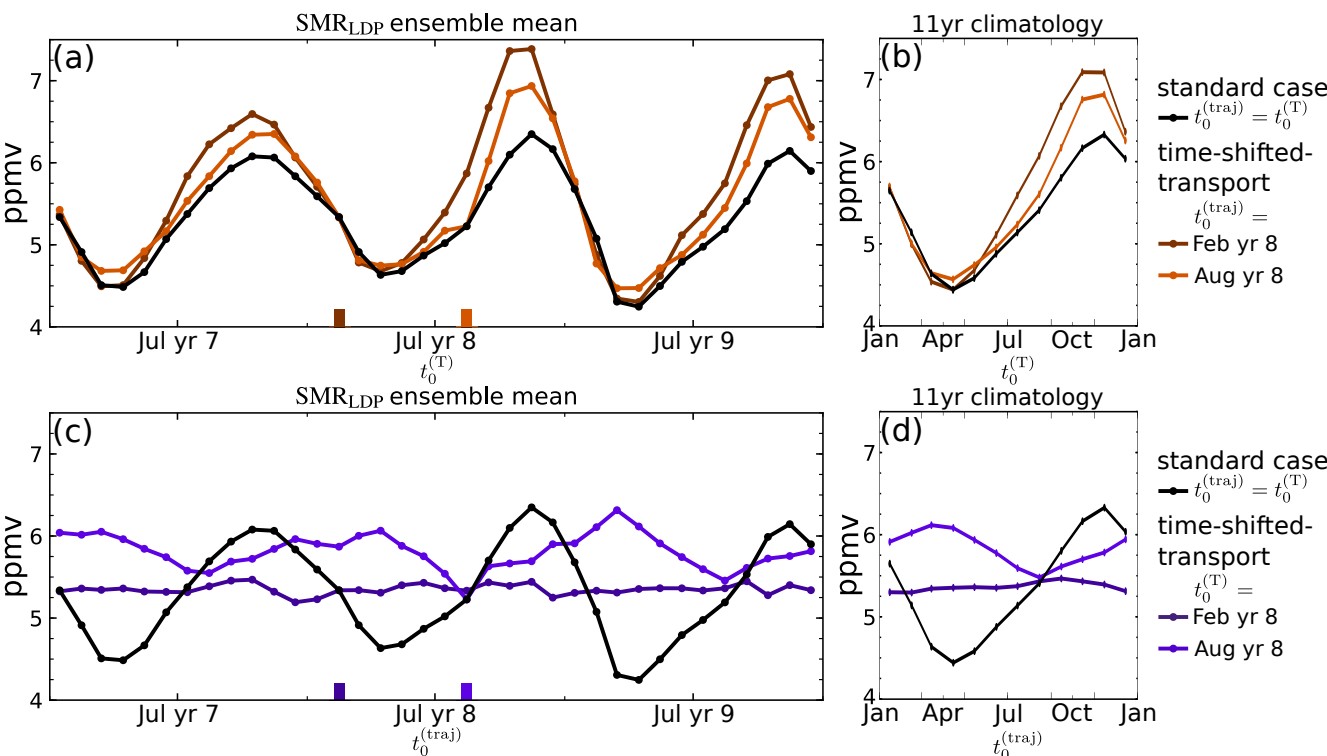

**Figure 12.** As for Fig. 6 but for the climate model. Three year timeseries of mean $SMR_{LDP}$ for trajectories of standard case (black line), (a) time-shifted-transport shifted to initialise in a single given month of year 8 (orange lines), (b) time-shifted-temperatures initialising in single given month of year 8 (purple lines).



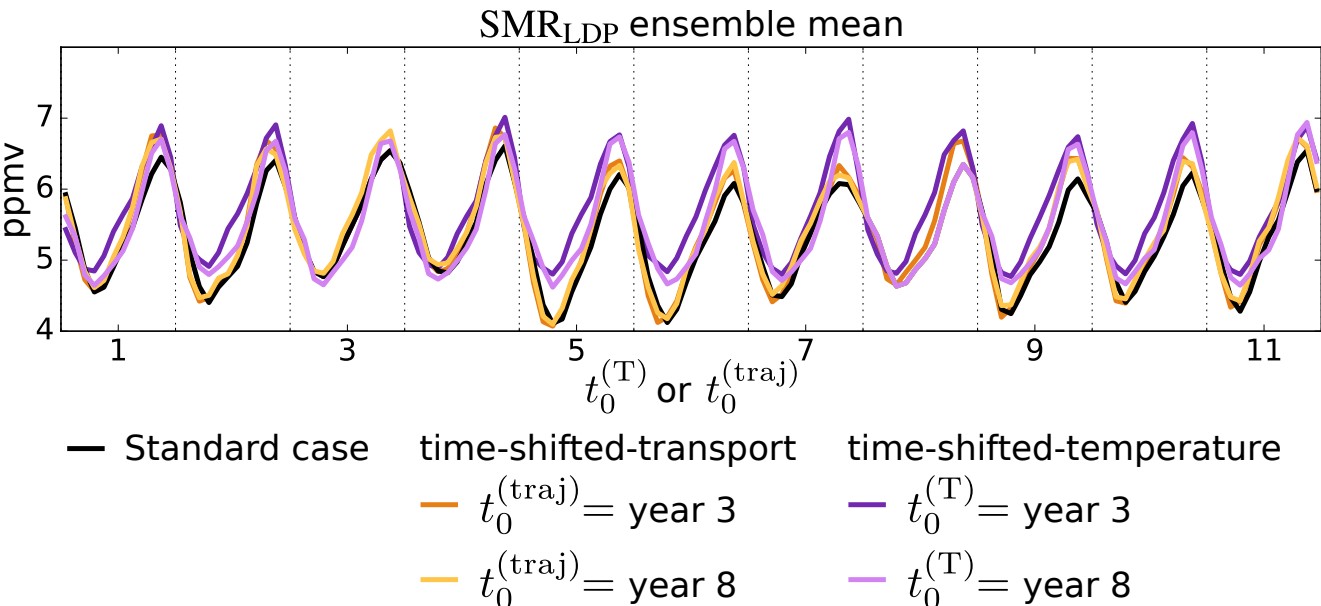

**Figure 13.** As for Fig. 3 but for the climate model. Timeseries of LDP calculations of water vapour entering the stratosphere for standard co-varying case (black line), time-shifted-transport fixed to initialise in a single given year (orange lines) and with time-shifted-temperatures fixed to initialise in a single given year (purple lines).



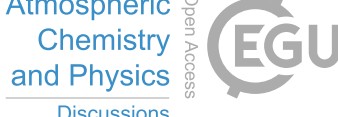

**Table 1.** Coefficient of determination ($R^2$) values for the standard and time-shifted cases in ERA-Interim over the period 1999-2009, and two cases from the climate model simulation. Values are calculated using monthly mean timeseries with the climatological annual cycle removed.

| $t_0^{(traj)}$ or $t_0^{(T)}$ | $R^2$ with standard case | |
| :---: | :---: | :---: |
| | time-shifted-transport | time-shifted-temperature |
| 1999 | 88 % | 6 % |
| 2000 | 71 % | 1 % |
| 2001 | 75 % | 13 % |
| 2002 | 72 % | 13 % |
| 2003 | 71 % | 0 % |
| 2004 | 74 % | 12 % |
| 2005 | 67 % | 1 % |
| 2006 | 70 % | 4 % |
| 2007 | 74 % | 2 % |
| 2008 | 88 % | 4 % |
| 2009 | 75 % | 1 % |
| climate model year 3 | 81 % | 1 % |
| climate model year 8 | 85 % | 3 % |