# Peer review of "Sensitivity of stratospheric water vapour to variability in tropical tropopause temperatures and large-scale transport"

_Atmospheric Chemistry and Physics, 2020_

## Referee Comment (RC1) · Anonymous Referee #1 · 23 Oct 2020

In their manuscript Smith et al. describe an analysis in which they separate the contribution of varying tropical tropopause temperatures and varying transport on the amount of water vapour entering the stratosphere. They apply the concept of Lagrangian Dry Point along trajectories using ERA-interim data for the time period 1999-2009 and generated model data from the chemistry-climate model UM-UKCA. To distinguish between the effect of TTL temperatures and transport on the amount of water vapour entering the stratosphere, they time-shift either the temperatures (keeping the trajectories for transport as they would be for the specific years) or time-shift the trajectories for transport (keeping the respective temperature cycles fixed). Smith et al. found out that the variation in tropopause temperature contributes to 70% of the observed interannual lower stratospheric water vapour variability and is the dominant driver for the annual cycle as well. Despite the important role of seasonal variations in tropopause temperature for the water vapour variability, transport contributes with 30 % in reducing the seasonal cycle maximum.

General comments: The paper is a very interesting and important contribution to the field of stratospheric water vapour. However, the text is sometimes difficult to follow and I don't understand what the text is aiming at in some parts. Moreover, I would suggest to shorten the text, focus on major results and omit results that do not contribute to the major findings. To understand the results, it is most important to understand the methods of time-temperature shift and time-transport shift. Here, however I had problems to reconcile the description in the text and Figure 1c. To understand your results, it is indispensable to understand the concept of time-shift methods, and this should be improved for the final paper. You divided your results chapter into the results of ERA-interim and model results. I suggest merging these chapters into one results section. This shortens the text and makes an inter-comparison easier. Moreover, the figures can be presented together.

The paper is suitable for publication in ACP after major revision.

Specific comments: Page 2, line 26: Please cite Stenke and Grewe (2005) here.

Stenke, A. and Grewe, V.: Simulation of stratospheric water vapor trends: impact on stratospheric ozone chemistry, Atmos. Chem. Phys., 5, 1257–1272, https://doi.org/10.5194/acp-5-1257-2005, 2005.

Chapter 2.1 and 2.2: Please include all aspects necessary for the description of the trajectory analysis in chapter 2.1. For instance: you say on page 5, line 131 (section 2.2), that you release the trajectories at the 83 hPa level. This information should be already available on page 4, line 96ff. It would also be nice to have an overview (table) over the important differences (period simulated, resolution of the data, release level of the trajectories ..) between the trajectory analysis for ERA and the model.

[Figure]

Page 4, line 97: Using 5580 trajectories for each initializing date means you use about 2 trajectories per grid point. Did you make a sensitivity test to show that your results are insensitive to that number? Please comment.

Page 4, line 106: Please highlight the formulae as an equation with a number.

Figure 3 b and d: I see the purple lines displayed as blue ones.

In figure 4, you switched the colours for the time-shifted transport from orange to blue (vice versa for the time-shifted temperature). This is a little confusing after reading figure 3.

Page 5, line 146: What do you mean by "pattern"? I would also suggest that there is a difference in the timing of the maxima in figure 2 between the SWOOSH data and the LDP calculation. Please comment.

Page 7, line 18: Please explain the term "generic" transport.

Page 8, line 250: Does "corresponding" mean that the coloured lines end at the month with the respective diamonds at 83 hPa? Please describe this more clearly.

Page 10, line 305: Please explain why the combination of initializing transport in August with temperatures in autumn leads to lower average SMR_LDP? From the text passage before, this is not clear to me.

Page 10, line 313 to 317: "For this temperature initialization date…SMR_LDF has a marked minimum, which appears consistent with the behavior of the time-shifted transport calculations discussed above." Please describe explicitly what is meant by "consistent with the behavior of the time-shifted transport calculations above."

Figure 7c: There is no structure visible in the figure. Please change selected range of the colour bar.

Page 11, line 47: Please specify what you mean by "any of the other features".

Page 11, line 58: Please specify what you mean by "cases".

Page 12, line 372: "which implies that cold points corresponding to LDPs will be cold relative…". I would suggest to write: "which implies that cold points corresponding to LDPs will be of very low temperatures relative.."

Page 12, line 393: "halving any arbitrarily chosen timescale results in an equivalent change in T_LDP and SMR_LDP". Please describe what is meant by "equivalent.

Page 13, line 403: Do you really mean that there is NO major change of LDPs over tropical America in figure 10? At least the change seems to be larger than over Africa.

Page 13, lines 404-405: You describe that the light blue symbols in Fig 9b are NOT re-evaluated from the original 6 hourly temperature calculations. This is, as far as I understand, in contradiction to what is stated on page 11, lines 359-360.

Page 13, line 405: "the difference between figure 10a and figure 10b is ignored"? Please explain in other words, what you wanted to say.

Page 13, line 405-415: It was hard to follow this text passage and there remain some questions: line 406: ".. does not require every trajectory to be search..". Why? What is meant by using "fixed LDPs". I looked through the text, but did not find a definition.

Page 13, line 419: How do you calculate backward trajectories when using zonally averaged temperatures?

Page 14, line 451: You state that vertical advection MAY be weaker in the model than in ERA-interim data. Why don't you look into the data and check if this is actually the case?

Page 14, line 460: You describe that the differences in saturation mixing ratios between model and reanalysis are difficult to explain. In line 455, however, you argue that the model trajectories probably do not sample the coldest regions of the TTL efficiently (as the ERA-interim trajectories do). As far as I understand, the last sentence is already

an explanation for the differences between ERA-data and model. Please comment.

Page 15, line 480: Do you mean "the seasonal variation in the fractional distributions" as indicated in figure 5 and 11 (lower panel i.e. 5c and 11c)?

Page 15, line 482: To which amplitude to you refer? This sentence is too long to understand, so please split it into 2 sentences of it and rephrase it

Page 16, line 518: What is meant by "without re-calculation the space-time positions of LDPs"?

Page 17, line 531: Please refer to the results section in which the importance of convective injection or particle formation or sedimentation were an outcome of your results.

Page 17, line 544: Please replace "cold temperatures" by "low temperatures".

Page 18, line 562: "The generally stronger...". The first part of the sentence is clear, but what does ".. consistent with the seasonal variation noted previously of the amount of interannual variability ..." mean? Please rephrase the text.

---

## Referee Comment (RC2) · Anonymous Referee #2 · 30 Oct 2020

**General:**
This a very important and well-written paper and should be published by ACP. It describes and uses the novel method of separation of transport and temperature histories on the formation of stratospheric water vapor. Although there are some limitations of this method (e.g. diabatic vertical velocities which are a part of transport strongly depend on temperatures), the obtained results are of great value. Similarly, the timescale-dependent analysis of Eulerian temperature variations give very interesting insights into their importance on stratospheric entry values of water vapor. Thus, I have only few minor comments.

[Figure]

**Major points**

- Maybe one additional sentence in the abstract stressing the importance of the sampling effect of the Lagrangian dry point reconstruction which can lead to so different values and patterns of water vapor entering the stratosphere if compared with Eulerian estimates...

**Minor comments:**

- P2L26
  ...impact of water vapor on ozone is also related to the impact on the formation of the polar stratospheric clouds (PSCs)...more PSCs, more catalytic ozone depletion... maybe you can mention it

- P4L122
  The sentence starting with "This value could be regarded..." is very difficult to understand and probably not necessary. Fig 1, its caption and the other parts of text explain fully sufficient the applied method...Maybe you can replace "hot" by "warm" in the color bar of Fig 1 (hot TTL sounds strange for me).

- P6L170
  In the description of the UM-UKCA simulation it is not clear for me how the inter-annual variability was realized in the time-slice simulation. Because of the perpetual year 2000 boundary condition, it is not clear if you run year 2000 many times (perpetuum run) and get in this way an ensemble of "different" years 2000 which mimic the inter-annual variability of the real atmosphere?

- P6L183
  The 11 orange lines are obtained by using the transport of each particular

year between 1999-2009 (11 cases) for all years between 1999 and 2009, isn't it?...maybe you would like to add this or a similar sentence to your text or to the caption of Fig 3a.

- P7L193-203
  To be honest I do not understand your explanation in this paragraph. For me every orange line in Fig 3a is calculated with always "true" temperature and "false" winds with exception of only one year when also the wind is correct (e.g. if you take the winds from 2003 for all other years between 1999 and 2009 than only for 2003 both temperature and winds are correct). Then, to get the purple line in Fig 4a you have to calculate the in-year average, i.e. an average over 10 orange lines from Fig 3a ....same with the orange line in Fig 4b resulting from 11 purple lines in Fig 3b. However, I do not understand why did you flip the colors? You also write in the caption of Fig 4 "....of (orange) time-shifted-temperatures and (purple) time-shifted-transport" Maybe you would like to clarify and reformulate this paragraph.

- P8L235
  The results presented in 3.1 are very interesting and important. One additional point: In Fig 4b the positive anomalies of the purple line are always between late spring and fall indicating the also the summer monsoons and their dynamical inter-annual variability may be an important factor...

- P8L249
  The dry bias and reduced annual cycle amplitude due to non-linearity of the Clausius-Clapeyron equation with respect to temperature...

- P9L255, Figure 5
  In the caption you should shortly denote the red dotted line as an Eularian estimation of the tropical H2O in the lower stratosphere..

- P9283-288
  Similar problem like in the previous chapter. I think that you repeat the transport of January for all month of a given year, the same for February, March, etc... and the same for temperature?

- P9L320
  Section 3.2 is also very interesting and has very valuable results. Still two remarks (1) In boreal summer the monsoon circulations are very strong and unique. I also expect some influence on your results if compared with the winter transport, i.e. meridional "wide" (summer) versus meridional "narrow" (winter) tropics. (2) Vertical velocities, i.e. diabatic heating rates you are using depend strongly on the lowest temperatures in the TTL, i.e. cold TTL is related to a strong upwelling (winter) and warm TTL is related to a weak upwelling (summer). Because of this, the separation between transport and temperature has a clear limitation...

- P13L395
  "Eulerian methods to estimate stratospheric water vapour"– in this context I can only imagine Eulerian methods to estimate temperature fluctuations which are compared here with the Lagrangian dry point estimation...how do you apply it for water vapor?... maybe you wish to clarify it

- P13L420
  You should mention here that the zonally averaged values are marked as stars in Fig 9.

- P14L433
  Why is the modeled tropical mean Eulerian water vapor (Fig 11 a, dotted line) higher than the H2O obtained from the Lagrangian reconstruction? Is the transport scheme of the chemistry-climate model too diffusive?

- P14L454

...for which the LDP in the first month or two - something wrong with the sentence

- P14L558
  "The seasonal variation of transport..." Do you mean the seasonal variation of vertical transport (different upwelling in winter and summer) or of horizontal transport (narrow tropics during winter and wider tropics including Asian and American monsoon during summer)?

- P18L564 The general stronger role of transport during boreal summer and fall can be due to the inter-annual variability of monsoons

---

## Author Comment (AC1) · 22 Dec 2020

Reviewer comment (RC): In their manuscript Smith et al. describe an analysis in which they separate the contri- bution of varying tropical tropopause temperatures and vary- ing transport on the amount of water vapour entering the stratosphere. They apply the concept of Lagrangian Dry Point along trajectories using ERA-interim data for the time period 1999-2009 and gen- erated model data from the chemistry-climate model UM-UKCA. To distinguish be- tween the effect of TTL temperatures and transport on the amount of water vapour entering the stratosphere, they time-shift either the temperatures (keeping the trajecto- ries for transport as they would be for the specific years) or time-shift the trajectories for transport (keeping the respective temperature cycles fixed). Smith et al. found out that the variation in tropopause temperature contributes to 70% of the observed inter- annual lower stratospheric water vapour variability and is the dominant driver for the annual cycle as well. Despite the important role of seasonal variations in tropopause temperature for the water vapour variability, transport contributes with 30 % in reducing the seasonal cycle maximum.

General comments: The paper is a very interesting and important contribution to the field of stratospheric water vapour. However, the text is sometimes difficult to follow and I don't understand what the text is aiming at in some parts. Moreover, I would suggest to shorten the text, focus on major results and omit results that do not contribute to the major findings. To understand the results, it is most important to understand the methods of time-temperature shift and time-transport shift. Here, however I had problems to reconcile the description in the text and Figure 1c. To understand your results, it is indispensable to understand the concept of time-shift methods, and this should be improved for the final paper. You divided your results chapter into the results of ERA-interim and model results. I suggest merging these chapters into one results section. This shortens the text and makes an inter-comparison easier. Moreover, the figures can be presented together.

Author comment (AC): We thank the reviewer for providing thorough and constructive comments. We agree it is crucial to introduce the methodology as clearly as possible. Therefore, we have rephrased the abstract as well as section 2.1. This has lead to the removal of figure 1. To simplify the terminology, we have replaced 'time-shifted temperatures' and 'time-shifted transport' with 'replaced-temperatures' and 'replaced-transport' throughout the text. Small changes have been made to the text throughout.

We have considered carefully the reviewer's suggestion to merge the results sections for ERA-Interim and for the chemistry-climate model. We have decided against that for two reasons. The first is that the figures are already busy and attempting to combine

figures will increase the density of information still further. The second is that the ERA-Interim and chemistry-climate models are not intended to be directly compared. The model has significant biases which have been clearly noted and the model simulation has not been designed to capture any aspect of actual historical interannual variability. The intention of including both ERA-Interim and model results in the paper is to demonstrate that the techniques we introduce can be usefully applied to each (and clearly in future investigations such techniques could be used for careful reanalysis-model comparisons, where those were appropriate). Given that the methods section is now clearer and shorter, we believe this suitably addresses the reviewer's key overall concern of clarity.

Responses to specific comments are below. Following the removal of figure 1, all other figures have been renumbered accordingly. The responses below will refer to the original figure numbers (where there are a total of 13 figures).

RC: The paper is suitable for publication in ACP after major revision. Specific comments:

Page 2, line 26: Please cite Stenke and Grewe (2005) here. Stenke, A. and Grewe, V.: Simulation of stratospheric water vapor trends: im- pact on stratospheric ozone chemistry, Atmos. Chem. Phys., 5, 1257–1272, https://doi.org/10.5194/acp-5-1257-2005, 2005.

AC: Thanks for pointing out this valuable reference. It is included in the revised version of the paper.

RC: Chapter 2.1 and 2.2: Please include all aspects necessary for the description of the trajectory analysis in chapter 2.1. For instance: you say on page 5, line 131 (section 2.2), that you release the trajectories at the 83 hPa level. This information should be already available on page 4, line 96ff. It would also be nice to have an overview (table) over the important differences (period simulated, resolution of the data, release level of the trajectories.) between the trajectory analysis for ERA and the model.

AC: After careful consideration, taking into account that direct intercomparison between model and re-analysis is not the intended aim of the paper we have kept the original structure and not include an intercomparison table. Section 2.1 is intended to describe the commonalities and the remaining details in section 2.2 and 2.3 are already referred to on page 4 line 96 ff. The paper is less interested in directly comparing LDPs resulting from ERA-Interim and UM-UKCA and more interested in whether the exhibited sensitivity is present in both.

RC: Page 4, line 97: Using 5580 trajectories for each initializing date means you use about 2 trajectories per grid point. Did you make a sensitivity test to show that your results are insensitive to that number? Please comment.

AC: We conducted a sensitivity test of the number of trajectories initialised at a single date, ranging from around 1000 to 44000. Results were insensitive for trajectory numbers spanning 5580 to 44000. A comment is made to reflect this in the text.

RC: Page 4, line 106: Please highlight the formulae as an equation with a number.

AC: The formula is now a numbered equation.

RC: Figure 3 b and d: I see the purple lines displayed as blue ones.

AC: Thanks for pointing out the ambiguity in this colour scheme. The orange and purple colours have been made more consistent between figures 3 and 4.

RC: In figure 4, you switched the colours for the time-shifted transport from orange to blue (vice versa for the time-shifted temperature). This is a little confusing after reading figure 3.

AC: P7L193-198 highlight the relationship that means the individual curves in figure 4 can be determined from either the orange or the blue lines of figure 3. Nevertheless, following the clarification of the method, those lines have been removed, so the colours in figure 4 have been switched and the legend rephrased accordingly.

[Figure]

RC: Page 5, line 146: What do you mean by "pattern"? I would also suggest that there is a difference in the timing of the maxima in figure 2 between the SWOOSH data and the LDP calculation. Please comment.

AC: Firstly, the word 'pattern' has been replaced with 'variability'. Secondly, there is a difference in the timing of the seasonal maxima for SWOOSH and SMR_LDP with ERA-I. A partial explanation is due to an error in generating the figure, in the timing of monthly datapoints. SWOOSH data is averaged monthly and displayed in the centre of the month, whereas trajectory results represent the first day of the month, but are incorrectly positioned in the centre of each month. For example, Figure 5 has the correct positioning. The horizontal positioning of the ERA-I SMR_LDP timeseries in Figure 2 has been corrected to reflect this. The vertical scale has also been changed to reduce the amount of white space at the top of the figure.

The difference in timing is therefore half a month, which is shorter than the frequency of trajectory initialisation. Further investigation would require a more frequent initialisation of trajectories.

Fueglistaler et al 2013 find good correlation between the same trajectories, with a slightly different LDP calculation, and the combination of satellite datasets HALOE and MLS/Aura. The main difference in the LDP calculation is a slightly different definition of troposphere-to-stratospheric transport. SWOOSH, the homogenised satellite observations of stratospheric water vapour used here, uses a different method to combine several satellite datasets including HALOE and MLS/Aura (Davis et al. 2016). The difference in seasonality here must be due to small differences in the methods of calculating LDPs and homogenising the satellite observations. We think that it is unneccesary to add this level of detail in the main text and is therefore not included in the revision.

RC: Page 7, line 218: Please explain the term "generic" transport.

AC: The unclear term 'generic transport' has been replaced with 'transport of alternative years'.

RC: Page 8, line 250: Does "corresponding" mean that the coloured lines end at the month with the respective diamonds at 83 hPa? Please describe this more clearly.

AC: This has been clarified in the text. Instead of : Figure 5b shows the seasonal variation of SMR_LDP, with each coloured line corresponding to a coloured diamond in Fig. 5a. It is now: Figure 5b shows the seasonal variation of SMR_LDP in the 12 month history from each initialisation date, with each coloured line corresponding to a coloured diamond in Fig. 5a.

RC: Page 10, line 305: Please explain why the combination of initializing transport in August with temperatures in autumn leads to lower average SMR_LDP? From the text passage before, this is not clear to me.

AC: This paragraph has been rephrased in an attempt to be clearer.

RC: Page 10, line 313 to 317: "For this temperature initialization date. . .SMR_LDF has a marked minimum, which appears consistent with the behavior of the time-shifted transport calculations discussed above." Please describe explicitly what is meant by "consistent with the behavior of the time-shifted transport calculations above."

AC: This sentence has been rephrased to: For this temperature initialisation date, the SMR_LDP has a marked minimum in July-October, with a similar explanation to the replaced-transport calculations discussed above. The following sentences provide explicit explanation.

RC: Figure 7c: There is no structure visible in the figure. Please change selected range of the colour bar.

AC: The ranges of the colour bars in Figure 7b and 7c have been narrowed, and the number of ticks has been increased. So as not to crowd their colour ranges, they now have different value ranges.

RC: Page 11, line 347: Please specify what you mean by "any of the other features".

AC: This sentenced has been rephrased to: The relative importance of temperature variability at different timescales for stratospheric water vapour depends on the effects on $\mathrm{SMR_{LDP}}$.

RC: Page 11, line 358: Please specify what you mean by "cases".

AC: 'Cases' has been replaced by 'T_LDP'

RC: Page 12, line 372: "which implies that cold points corresponding to LDPs will be cold relative. . .". I would suggest to write: "which implies that cold points corresponding to LDPs will be of very low temperatures relative.."

AC: Rewritten accordingly.

RC: Page 12, line 393: "halving any arbitrarily chosen timescale results in an equivalent change in T_LDP and SMR_LDP". Please describe what is meant by "equivalent.

AC: 'Equivalent change' has been replaced by 'equivalent reduction'.

RC: Page 13, line 403: Do you really mean that there is NO major change of LDPs over tropical America in figure 10? At least the change seems to be larger than over Africa.

AC: The black contour lines in Figure 10 suggest a change in the more confined area to the north of South America, but the intention of the sentence is to describe the redistribution of LDPs on the global tropical scale. The sentence has been rewritten to refer to the percentages in the dashed boxes. The dashed boxes suggest that LDPs redistribution to the West Pacific and SE Asia is not at the expense of the LDPs over Africa and America.

RC: Page 13, lines 404-405: You describe that the light blue symbols in Fig 9b are NOT re-evaluated from the original 6 hourly temperature calculations. This is, as far as I understand, in contradiction to what is stated on page 11, lines 359-360.

AC: Apologies, the sentence on lines 359-360 was confusing, it has been rephrased to

make clear that what is not re-evaluated is the location.

RC: Page 13, line 405: "the difference between figure 10a and figure 10b is ignored"? Please explain in other words, what you wanted to say.

AC: Thanks for pointing out this unclear statement, it is now described more explicitly.

RC: Page 13, line 405-415: It was hard to follow this text passage and there remain some questions: line 406: ".. does not require every trajectory to be search..". Why? What is meant by using "fixed LDPs". I looked through the text, but did not find a definition.

AC: The first sentences of this paragraph have been rewritten to introduce the term 'fixed-location LDPs'.

RC: Page 13, line 419: How do you calculate backward trajectories when using zonally averaged temperatures?

AC: Back-trajectories are calculated as before, their location in space and time is recorded according to the velocity field they encounter. Zonally averaged temperature is recorded as a passive variable. SMR is then calculated along the trajectory's record of pressure and averaged temperature, and the minimum SMR value defines the LDP in this case. The sentence has been rephrased to make this clearer.

RC: Page 14, line 451: You state that vertical advection MAY be weaker in the model than in ERA-interim data. Why don't you look into the data and check if this is actually the case?

AC: Various measures not considered in this paper indicate that vertical transport is weaker in the model than in ERA-Interim. These measures include transport time from troposphere to stratosphere, and Transformed Eulerian Mean vertical velocity. The residual vertical velocity at 70hPa and averaged across 30 N-S and averaged across the years analysed in the paper: - UM-UKCA is 2.1 mm/s - ERA-I is 2.3 mm/s which suggests a weaker vertical upwelling in the climate model. The text has been revised

accordingly.

RC: Page 14, line 460: You describe that the differences in saturation mixing ratios between model and reanalysis are difficult to explain. In line 455, however, you argue that the model trajectories probably do not sample the coldest regions of the TTL efficiently (as the ERA-interim trajectories do). As far as I understand, the last sentence is already an explanation for the differences between ERA-data and model. Please comment.

AC: The difficulty is related to a subtlety in the temperature sampling: What is happening to cause trajectories to bypass the lower temperatures in the TTL? For the climate model, it occurs less often but experiences a more extreme SMR. Is the cause some difference in vigorous vertical transport due to the choice of kinematic advection scheme, or some model-related representation of advection near the tropopause, either numerical or physical, such as vertical interpolation or deep convection? As your major point is to focus the text, this sentence has been removed.

RC: Page 15, line 480: Do you mean "the seasonal variation in the fractional distributions" as indicated in figure 5 and 11 (lower panel i.e. 5c and 11c)?

AC: Thanks, this has been clarified.

RC: Page 15, line 482: To which amplitude to you refer? This sentence is too long to understand, so please split it into 2 sentences of it and rephrase it

AC: This has been rephrased accordingly.

RC: Page 16, line 518: What is meant by "without re-calculation the space-time positions of LDPs"?

AC: This has been clarified.

RC: Page 17, line 531: Please refer to the results section in which the importance of con- vective injection or particle formation or sedimentation were an outcome of your

results.

AC: This has been rephrased. Our argument is that the results of this paper contribute to the overall problem of understanding the relative importance of the several different processes that are needed to explain swv variability. But certainly not all such processes, which include convective injection and particle formation, have been investigated here.

RC: Page 17, line 544: Please replace "cold temperatures" by "low temperatures".

AC: Thanks, this has been replaced.

RC: Page 18, line 562: "The generally stronger. . .". The first part of the sentence is clear, but what does ".. consistent with the seasonal variation noted previously of the amount of interannual variability . . ." mean? Please rephrase the text.

AC: This has been clarified.

---

## Author Comment (AC2) · 22 Dec 2020

Reviewer comment (RC): General: This a very important and well-written paper and should be published by ACP. It de- scribes and uses the novel method of separation of transport and temperature histories on the formation of stratospheric water vapor. Although there are some limitations of this method (e.g. diabatic vertical velocities which are a part of transport strongly de- pend on temperatures), the obtained results are of great value. Similarly, the timescale- dependent analysis of Eulerian temperature variations give very interesting insights into their importance on stratospheric entry

values of water vapor. Thus, I have only few minor comments.

Author comment (AC): We thank the reviewer for their supportive and insightful comments which have improved the paper. Responses to specific comments are below. Small changes have been made to the text throughout.

Regarding responses: As a result of revising sections 2.1 and 3.1 for reviewer 1, figure 1 is now removed. So all figures 2,3...13 are instead number 1,2,...12 in the revised manuscript. All of the figure numbers mentioned below refer to the initial submission.

RC: Major points

Maybe one additional sentence in the abstract stressing the importance of the sampling effect of the Lagrangian dry point reconstruction which can lead to so different values and patterns of water vapor entering the stratosphere if compared with Eulerian estimates. . .

AC: This sentence has been added: "As with other aspects of dehydration, simple Eulerian measures of variability are not sufficient to quantify the implications for dehydration and the Lagrangian sampling of the variability must be taken into account."

RC: Minor comments:

P2L26 ...impact of water vapor on ozone is also related to the impact on the formation of the polar stratospheric clouds (PSCs)...more PSCs, more catalytic ozone de- pletion... maybe you can mention it

AC: PSCs are now mentioned.

RC: P4L122 The sentence starting with "This value could be regarded..." is very difficult to understand and probably not necessary. Fig 1, its caption and the other parts of text explain fully sufficient the applied method...Maybe you can replace "hot" by "warm" in the color bar of Fig 1 (hot TTL sounds strange for me).

AC: To address this and the comments of reviewer 1, this paragraph and figure 1 have been removed. Rather than referring to two timeseries, the method is presented as sensitivity experiment with replacement temperatures or replacement transport.

RC: P6L170 In the description of the UM-UKCA simulation it is not clear for me how the inter-annual variability was realized in the time-slice simulation. Because of the perpetual year 2000 boundary condition, it is not clear if you run year 2000 many times (perpetuum run) and get in this way an ensemble of "different" years 2000 which mimic the inter-annual variability of the real atmosphere?

AC: A sentence has been added to this paragraph to clarify what type of interannual variability is present in UM-UKCA. Also, to reflect the final figures presented in this paper, the statement that '49 years of data are used to calculate back trajectories' has been corrected to 12 years.

RC: P6L183 The 11 orange lines are obtained by using the transport of each particular year between 1999-2009 (11 cases) for all years between 1999 and 2009, isn't it?...maybe you would like to add this or a similar sentence to your text or to the caption of Fig 3a.

AC: Thanks, a more explicit description has been added to the caption of Fig3a.

RC: P7L193-203 To be honest I do not understand your explanation in this paragraph. For me every orange line in Fig 3a is calculated with always "true" temperature and "false" winds with exception of only one year when also the wind is correct (e.g. if you take the winds from 2003 for all other years between 1999 and 2009 than only for 2003 both temperature and winds are correct). Then, to get the purple line in Fig 4a you have to calculate the in-year average, i.e. an average over 10 orange lines from Fig 3a ....same with the orange line in Fig 4b resulting from 11 purple lines in Fig 3b. However, I do not understand why did you flip the colors? You also write in the caption of Fig 4 "....of (orange) time-shifted-temperatures and (purple) time-shifted-transport" Maybe you would like to clarify and reformulate this paragraph.

AC: Also in response to reviewer 1, this paragraph and related descriptions in the abstract and methods section have been reformulated to be more understandable. The phrases 'time-shifted temperatures' and 'time-shifted transport' have been replaced with 'replaced temperatures' and 'replaced transport' and described as sensitivity experiments. As part of reformulating the text, the colours in figure 4 have also been swapped. The description of the relationship between purple and orange lines has been removed as it is a subtlety that obscures the main results.

RC: P8L235 The results presented in 3.1 are very interesting and important. One additional point: In Fig 4b the positive anomalies of the purple line are always between late spring and fall indicating the also the summer monsoons and their dynamical inter-annual variability may be an important factor...

AC: Thanks, this is an important point across sections 3.1 and 3.2. It therefore included in the discussion.

RC: P8L249 The dry bias and reduced annual cycle amplitude due to non-linearity of the Clausius-Clapeyron equation with respect to temperature...

AC: This has been included.

RC: P9L255, Figure 5 In the caption you should shortly denote the red dotted line as an Eulerian esti- mation of the tropical H2O in the lower stratosphere..

AC: This has been rephrased to be clearer.

PHH: change 'a Eulerian estimate' to 'an Eulerian estimate'.

RC: P9283-288 Similar problem like in the previous chapter. I think that you repeat the transport of January for all month of a given year, the same for February, March, etc... and the same for temperature?

AC: This paragraph has been rephrased accordingly.

RC: P9L320 Section 3.2 is also very interesting and has very valuable results. Still two

re- marks (1) In boreal summer the monsoon circulations are very strong and unique. I also expect some influence on your results if compared with the winter transport, i.e. meridional "wide" (summer) versus meridional "narrow" (winter) tropics. (2) Vertical velocities, i.e. diabatic heating rates you are using depend strongly on the lowest temperatures in the TTL, i.e. cold TTL is related to a strong upwelling (winter) and warm TTL is related to a weak upwelling (summer). Because of this, the separation between transport and temperature has a clear limitation...

AC: The influence of tropical width will be part of the results described for Fig 6. It would be interesting to view the horizontal distribution of LDP events but is not presented here. For a limited view, see my thesis, Smith (2020) Fig 4.13, which may suggest that the summers of anomalous efficiency (1999, 2008) are connected to summers with a broader tropical region near the maritime continent. This is an interesting topic beyond the scope of this study but could be investigated in a further piece of work.

Such a separation between temperature and transport is inevitably artificial because in reality aspects of temperature and transport are to some extent coupled, but useful insight emerges. To reflect this, a sentence has been included to reflect this in section 2.1 and the second paragraph of section 5.

RC: P13L395 "Eulerian methods to estimate stratospheric water vapour"– in this context I can only imagine Eulerian methods to estimate temperature fluctuations which are compared here with the Lagrangian dry point estimation...how do you apply it for water vapor?... maybe you wish to clarify it

AC: This sentence was unclear and has been made more explicit.

RC: P13L420 You should mention here that the zonally averaged values are marked as stars in Fig 9.

AC: This has been included.

RC: P14L433 Why is the modeled tropical mean Eulerian water vapor (Fig 11 a, dotted

line) higher than the $H_2O$ obtained from the Lagrangian reconstruction? Is the transport scheme of the chemistry-climate model too diffusive?

AC: It could be because of differences in their respective transport schemes, such as diffusive climate model transport, or processes missing from the simple LDP calculation such as microphysics. This requires a careful investigation of the model transport scheme and we prefer not to speculate on this, so a general comment has been made in the first paragraph of section 4.1.

RC: P14L454 ...for which the LDP in the first month or two - something wrong with the sentence

AC: This sentence has been clarified.

RC: P14L558 "The seasonal variation of transport..." Do you mean the seasonal variation of vertical transport (different upwelling in winter and summer) or of horizontal trans- port (narrow tropics during winter and wider tropics including Asian and American monsoon during summer)?

AC: In the form of these sensitivity experiments, transport is general and referrs to both vertical and horizontal directions. This is now noted in section 3.1, and repeated in the summary section.

RC: P18L564 The general stronger role of transport during boreal summer and fall can be due to the inter-annual variability of monsoons

AC: This is now highlighted earlier in the paragraph, which helps to point out where vertical and horizontal aspects transport are considered.